# Predicting precision grip grasp locations on three-dimensional objects

**Lina K. Klein**[1☉‡], **Guido Maiello**[1☉‡*], **Vivian C. Paulun**[1], **Roland W. Fleming**[1,2]

**1** Department of Experimental Psychology, Justus Liebig University Giessen, Giessen, Germany, **2** Center for Mind, Brain and Behavior, Justus Liebig University Giessen, Giessen, Germany

☉ These authors contributed equally to this work.
‡ These authors share first authorship on this work
* guido_maiello@yahoo.it

**Data Availability Statement:** Data and analysis scripts as well as supplementary figures are available from the Zenodo database (doi:10.5281/zenodo.3891663).

## Abstract

We rarely experience difficulty picking up objects, yet of all potential contact points on the surface, only a small proportion yield effective grasps. Here, we present extensive behavioral data alongside a normative model that correctly predicts human precision grasping of unfamiliar 3D objects. We tracked participants' forefinger and thumb as they picked up objects of 10 wood and brass cubes configured to tease apart effects of shape, weight, orientation, and mass distribution. Grasps were highly systematic and consistent across repetitions and participants. We employed these data to construct a model which combines five cost functions related to force closure, torque, natural grasp axis, grasp aperture, and visibility. Even without free parameters, the model predicts individual grasps almost as well as different individuals predict one another's, but fitting weights reveals the relative importance of the different constraints. The model also accurately predicts human grasps on novel 3D-printed objects with more naturalistic geometries and is robust to perturbations in its key parameters. Together, the findings provide a unified account of how we successfully grasp objects of different 3D shape, orientation, mass, and mass distribution.

## Author summary

A model based on extensive behavioral data unifies the varied and fragmented literature on human grasp selection by correctly predicting human grasps across a wide variety of conditions.

## Introduction

In everyday life, we effortlessly grasp and pick up objects without much thought. However, this ease belies the computational complexity of human grasping. Even state of the art robotic AIs fail to grip objects nearly 20% of the time [1]. To pick something up, our brains must work out which surface locations will lead to stable, comfortable grasps, so we can perform desired actions (Fig 1A). Most potential grasps would actually be unsuccessful, e.g., requiring thumb

**Funding:** This research was supported by the DFG (IRTG-1901: 'The Brain in Action', SFB-TRR-135: 'Cardinal Mechanisms of Perception', and project PA 3723/1-1), and an ERC Consolidator Award (ERC-2015-CoG-682859: 'SHAPE'). Guido Maiello was supported by a Marie-Skłodowska-Curie Actions Individual Fellowship (H2020-MSCA-IF-2017: 'VisualGrasping' Project ID: 793660). The funders had no role in study design, data collection and analysis, decision to publish, or preparation of the manuscript.

**Competing interests:** The authors have declared that no competing interests exist.

and forefinger to cross, or failing to exert useful forces (Fig 1B). Even many possible grasps would be unstable, e.g., too far from the object's center, so it rotates when lifted (Fig 1C). Somehow, the brain must infer which, of all potential grasps, would actually succeed. Despite this, we rarely drop objects or find ourselves unable to complete actions because we are holding them inappropriately. How does the brain select stable, comfortable grasps onto arbitrary 3D objects, particularly objects we have never seen before?

Despite the extensive literature describing human grasping patterns, movement kinematics, and grip force adjustments [2–14], little is understood about the computational basis of initial grasp selection. Few authors have attempted to study and model how humans select grasps (e.g. [15,16]), and even then, only for 2D shapes. This is because, even for two-digit precision grip, many factors influence grasping. Object shape must be considered, since the surface normals at contact locations must be approximately aligned (a concept known as force closure [17]), otherwise the object will slip through our fingertips (Fig 1B, bottom). Object mass and mass distribution must be evaluated, since for grips with high torques (i.e. far from the center of mass, CoM [18–22]) the object will tend to rotate under gravity and potentially slip out of our grasp (Fig 1C, top). The orientation [19,22–25] and size [26] of grasps on an object must be considered, since the arm and hand can move and apply forces only in specific ways. Grasps that do not conform to the natural configuration of our hand in 3D space might be impossible (Fig 1B, top), or uncomfortable (Fig 1C, bottom). The hand's positioning may also determine an object's visibility [9, 27–30].

Most previous research did not assess the relative importance of these factors, nor how they interact. Here we sought to unify these varied and fragmented findings into a single normative framework. We therefore constructed a rich dataset in which we could tease apart how an object's 3D shape, mass, mass distribution, and orientation influence grasp selection. We devised a set of objects made of wood and brass cubes in various configurations (Fig 2), and asked participants to pick them up with a precision grip, move them a short distance and place

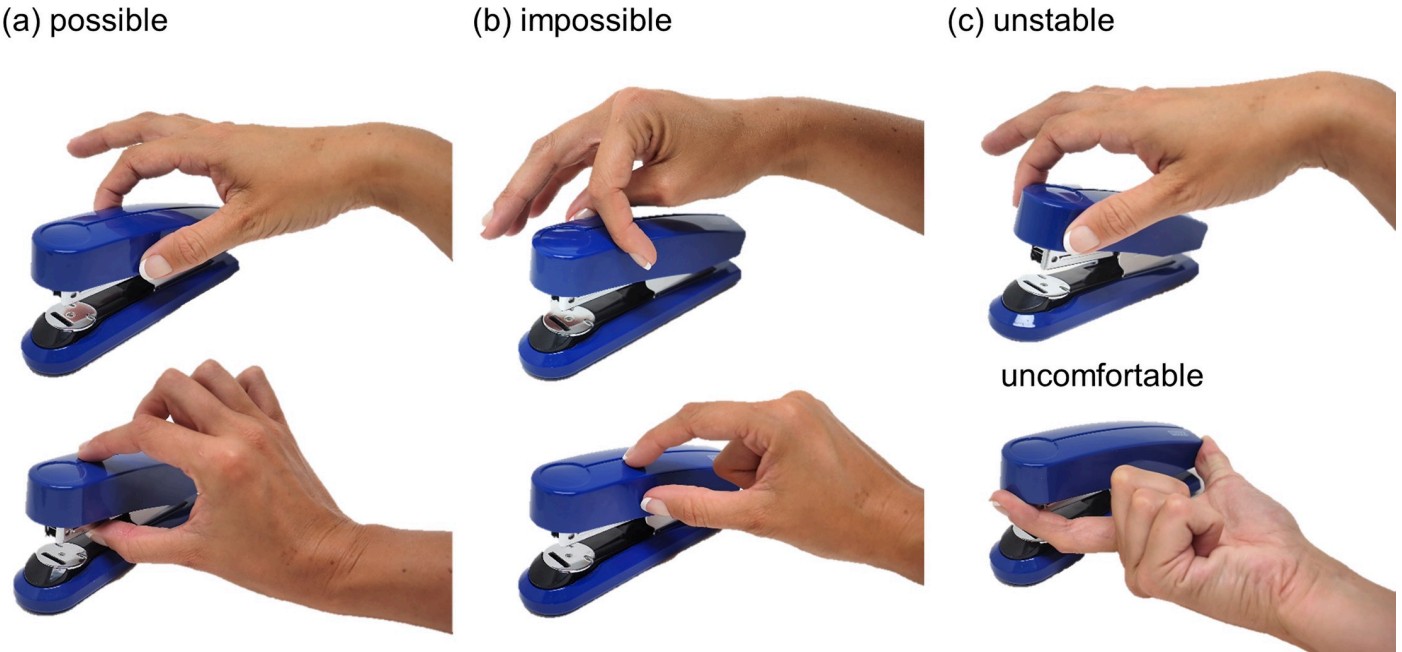

**Fig 1. The computational complexity of human grasp selection.** (a) Possible (b) Impossible (c) Possible but uncomfortable or unstable grasps.

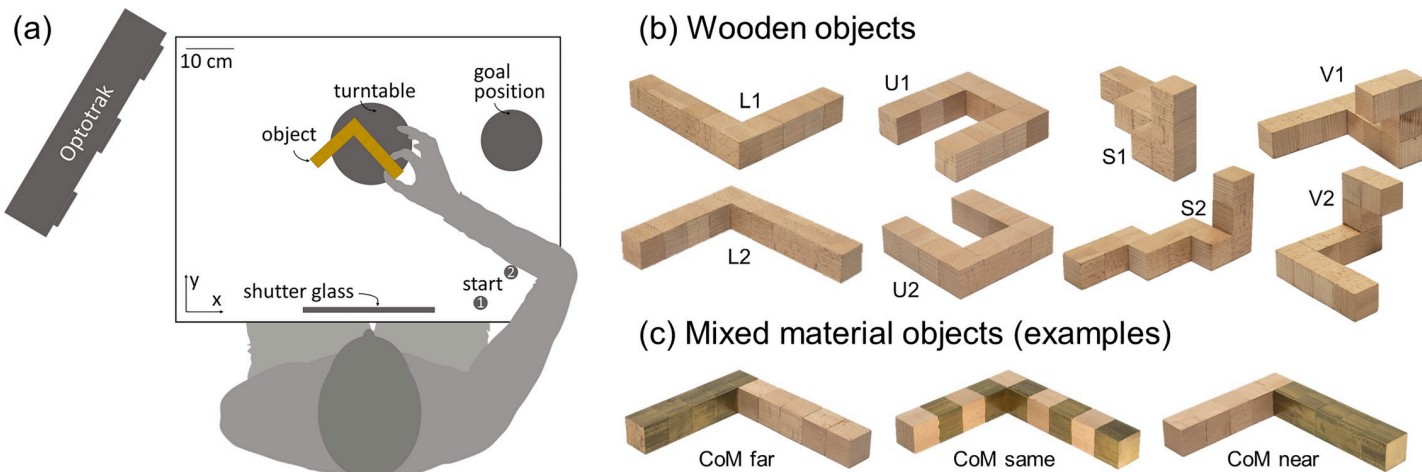

**Fig 2. Setup and stimuli for Experiments 1 and 2.** (a) Experimental setup. Seated participants performed grasping movements with their right hand. Following an auditory signal (coinciding with the shutter window turning transparent) they moved from one of the starting positions to the object and grasped it with a precision grip. They transported and released the object at the goal position and returned to the start position. (b) In Experiment 1 we employed four objects made of wooden cubes. Each object had a unique shape (that here we name L, U, S, V) and was presented at one of two different orientations with respect to the participant. (c) In Experiment 2 the objects had the same shapes as in Experiment 1, but now were made of wood and brass cubes. The brass and wood cubes were organized either in an alternate pattern (middle), so that the CoM of the object would remain approximately the same as for the wooden object, or grouped so that the CoM would be shifted either closer to (right) or away from (left) the participant's hand starting location.

them at a target location, while we tracked their thumb and forefinger. We measured initial contact locations (i.e. not readjusted contact regions during movement execution). By varying the shapes and orientation of the objects in Experiment 1, we (i) determined how consistent at selecting grasp locations participants are with themselves and other people, and (ii) measured the interactions between allocentric 3D shape and egocentric perspective on those shapes. If actors take the properties of their own effectors into account (e.g., hand orientation, grasp size), we should expect the same shape to be grasped at different locations depending on its orientation relative to the observer [19]. In Experiment 2, we varied the mass and mass distribution of the objects (Fig 2C) to test the relative role of 3D shape and mass properties. If participants take torques into account, identical shapes with different mass distributions should yield systematically different grasps [18,20–22].

Next, we employed this rich dataset to develop a computational model to predict human grasp patterns. We reasoned that grasps are selected to minimize costs associated with instability and discomfort. Accordingly, we implemented a model that combines five factors computed from the object's shape, mass distribution, and orientation: (i) force closure [17], (ii) torque [18–22] (iii) natural grasp axis [19,23–25], (iv) natural grasp aperture for precision grip [26] and (v) visibility [27,28]. The model takes as input a near-veridical 3D mesh representation of on object to be grasped, performs free-body computations on the mesh, and outputs minimum-cost, optimal grasp locations on the object. We found that the optimal grasps predicted by the model matched human grasp patterns on the wooden and brass polycube objects from Experiments 1 and 2 strikingly well. We then employed the model to generate predictions regarding where humans should grasp novel shapes with curved surfaces. In a final Experiment 3, we had participants grasp these novel 3D-printed, curved, plastic objects. Human grasps well aligned with the model predictions. Finally, we employed these data to show that model predictions are robust to perturbations in the model input and key parameters.

## Results

### Experiment 1: 3D shape and orientation

**Human grasps are tightly clustered and represent a highly constrained sample from the space of potential grasps.**   Twelve participants grasped four objects made of beech wood presented at two orientations (Fig 2A and 2B; see Materials and methods). Fig 3A shows how grasp patterns tend to be highly clustered. In each condition, different grasps have similar sizes (finger-to-thumb distance) and orientations, and also cover the same portions of the objects. Fitting multivariate Gaussian mixture models to the responses reveals that grasps cluster around only 1, 2, or 3 modes. Fig 3B shows three distinct modes for object U at orientation 2 in a unitless 2D representation of grasp space. Human grasps cover only a minute portion of the space of potential grasps. Note that we define the space of potential grasps as the set of all combinations of thumb and index finger positioning attemptable on the accessible surfaces of an object (i.e., those not in contact with the table). Fig 3C also shows how, for one representative condition, different grasps from the same subjects are more clustered than grasps from different subjects, since individuals predominantly selected only one (70%) or two (27%) modes, and only rarely (3%) grasped objects in three separate locations.

To further quantify how clustered these grasping patterns are we designed a simple metric of similarity between grasps (see Materials and methods). Fig 3D shows how both between- and within-subject grasp similarity are significantly higher than the similarity between random grasps only constrained by accessible object geometry (t(7)=9.96, p=2.2*10⁻⁵ and t(7)=26.15,

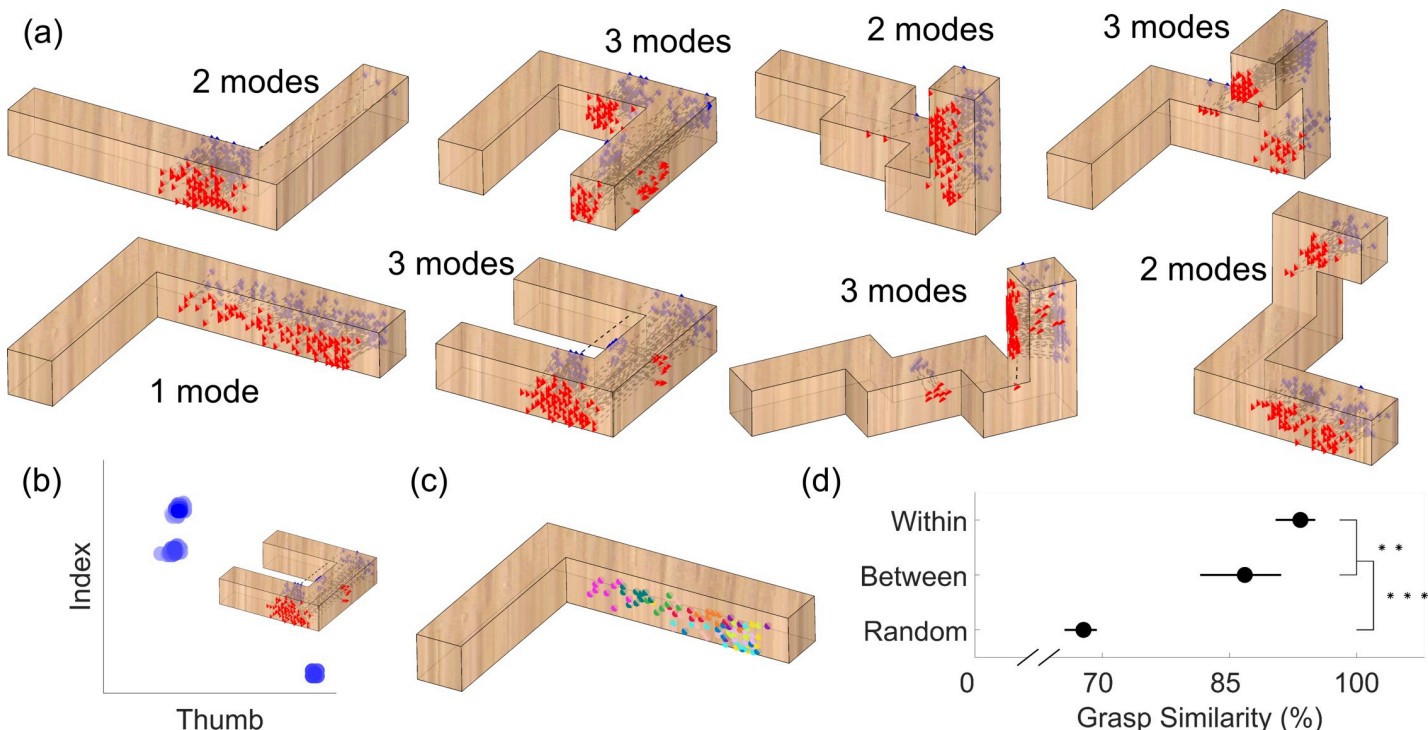

**Fig 3. Human grasps are clustered.** (a) Human grasps from Experiment 1. Grasps are represented as thumb (red triangles) and index finger (blue diamonds) contact positions, connected by dotted black lines. (b) Human grasps (blue blobs) for object U, orientation 2, when projected in a unitless 2D representation of the space of potential grasps, cluster around three distinct modes. (c) Distribution of thumb contact points on object L, orientation 2. Different colors represent grasps from different participants. (d) The level (%) of grasp similarity expected for grasps randomly distributed on the object surface (i.e. random combinations of thumb and index finger positioning attemptable on an object) and the observed level of between- and within-participant grasp similarity, averaged across objects and orientations. Error bars are 95% bootstrapped confidence intervals of the mean. ** p<0.01, *** p<0.001.

p=3.1*10^{-8} respectively). Additionally, within-subject grasp similarity is significantly higher than between subjects (t(7)=3.89, p=0.0060). Nevertheless, the high similarity between grasps from different participants demonstrates that different individuals tend to grasp objects in similar ways. The even higher level of within-subject grasp similarity further demonstrates that grasp patterns from individual participants are idiosyncratic, which may reflect differences in the strategies employed by individual participants, or may be related to physiological differences in hand size, strength, or skin slipperiness. We observe no obvious learning effects across trial repetitions: between-subject grasp similarity does not change from first to last repetition across objects and orientations (t(7)=0.62, p=0.56).

**Findings reproduce several known effects in grasp selection.** Previous research suggests haptic space is encoded in both egocentric and allocentric coordinates [31], and that grasps are at least partly encoded in egocentric coordinates to account for the biomechanical constraints of our arm and hand [19]. Our findings reproduce and extend these observations. If humans selected grasps in allocentric coordinates tied to an object's 3D shape, then grasps onto the same object in different orientations should be located on the same portions of the object but in different 3D world coordinates. Conversely, if actors take their own effectors into account, they should grasp objects at different locations depending on the object's orientation. For each object we computed grasp similarity across the two orientations in both egocentric (tied to the observer) and allocentric coordinates (tied to the object). Fig 4A shows that, as the extent of the object rotation increases, grasp encoding shifts from allocentric to egocentric coordinates. Across small rotations (object S, 55 degree rotation), grasps are more similar if encoded in allocentric coordinates (t(11)=13.90, p=2.5*10^{-8}), whereas for large rotations (object L, 180 degrees) grasps are more similar if encoded in egocentric coordinates (t(11)=4.59, p= 7.8*10^{-4}). Therefore, both 3D shape as well as movement constraints influence grasps.

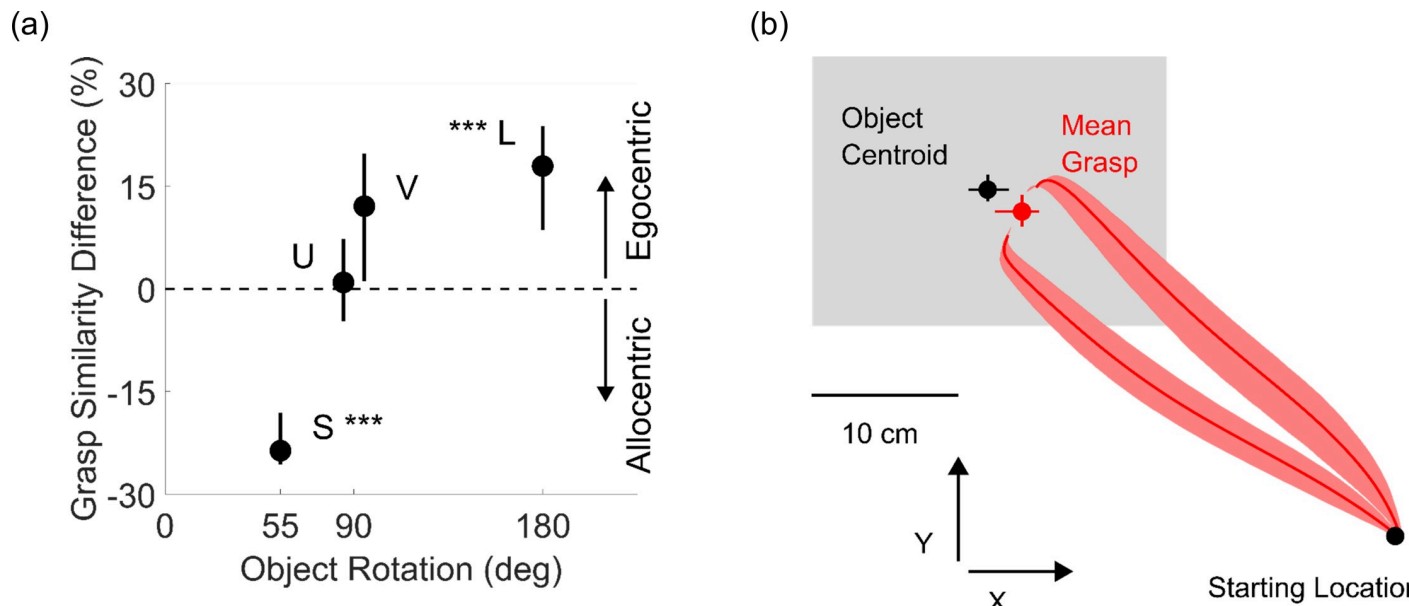

**Fig 4. Spatial encoding and bias.** (a) Difference in grasp similarity across orientations when grasps were encoded in object-centered (allocentric) vs human-centered (egocentric) coordinates, as a function of magnitude of rotation across the two orientation conditions. (b) Average grasp trajectories viewed in the x-y plane (red curves) from start location towards the objects (always contained within the gray shaded region). The average human grasp (red dot) across conditions is biased toward shorter reaching movements compared to the object centroids (black dot). In both panels data are means, error bars/regions represent 95% bootstrapped confidence intervals. *** p<0.001.

Fig 4B shows that participants also selected grasps that were on average 26 mm closer to the starting location than the object centroid (t(11)=9.74, p=9.6*10⁻⁷), reproducing known spatial biases in human grasp selection [15, 28, 30, 32, 33].

Consistent with Kleinholdermann et al. [15] but contrary to previous claims [18–22], our findings suggest humans care little about torque when grasping lightweight objects (of ~100 g). If actors sought to minimize torque, the selected grasps should be as close as possible to the CoM. Conversely, if participants were to disregard torque, then grasps should be at least as distant from the CoM as grasps randomly selected on the surface of the object. Fig 5A plots the difference between the CoM distance of participant grasps and the average CoM distance of random grasps, which we name 'CoM attraction compared to random grasps'. In Experiment 1, grasps were on average 9 mm farther from the CoM than the average distance to the object's CoM of grasps uniformly sampled onto the surface of the objects (t(11)=4.53, p=8.6*10⁻⁴). This negative value means that participants grasped the objects towards their extremities, farther from the CoM than even random chance.

## Experiment 2: Mass and mass distribution

**Humans grasp objects close to their center of mass when high grip torques are possible and instructions demand the object does not rotate.**   Due to the low density of beech wood, even the grasps farthest from the CoM in Experiment 1 would produce relatively low torques. Therefore, in Experiment 2 we tested whether participants grasp objects closer to the CoM when higher torques are possible. We did this by using objects of greater mass and asymmetric mass distributions. Specifically, for each of the shapes in Experiment 1, we made three new objects, each made of five brass and five wooden cubes: two 'bipartite' objects, with brass clustered on one or the other half of the object, and one 'alternating' object, with brass and wood alternating along the object's length. These objects had the same 3D shapes as in Experiment 1, but were nearly tenfold heavier (Fig 2C, see Materials and methods).

Fig 5A shows how human grasps are indeed significantly attracted towards the CoM of heavy objects, presumably to counteract the larger torques associated with higher mass. In Experiment 2, grasps were on average 11 mm closer to the object CoM than grasps sampled uniformly from the objects' surfaces (t(13)=4.89, p= 2.9*10⁻⁴), and on average 20 mm closer than the grasps from Experiment 1 (t(24)=6.60, p= 8.0*10⁻⁷). Fig 5B shows how this behavior was evident already from the very first trial performed by participants, but also that grasps clustered more toward the object CoM in later trials, presumably as participants refined their estimates of CoM location (correlation between CoM attraction and trial repetition: r = 0.86, p = 0.13). Importantly, participants shifted their grasps towards the CoM—not the geometrical centroid—of the objects (observe how the grasp patterns shift in Fig 5C). Fig 5D shows that when the object CoM was shifted towards the hand's starting location, participants did not significantly adjust their grasping strategy compared to Experiment 1 (t(13)=0.81, p=0.43). Conversely, when the object CoM was in the same position as in Experiment 1, grasps shifted on average by 8 mm towards the CoM (t(13)=3.92, p=0.0017). When the CoM was shifted away from the hand's starting position, grasps were on average 37 mm closer to the CoM compared to Experiment 1 (t(13)=8.49, p=1.2*10⁻⁶), a significantly greater shift than both the near and same CoM conditions (t(13)=8.66, p=9.2*10⁻⁷ and t(13)=7.58, p=4.0*10⁻⁶). These differential shifts indicate that participants explicitly estimated each object's CoM from visual material cues.

Even with the heavier objects, participants still systematically selected grasps that were closer to the starting location than the object centroid (t(13)=4.03, p=0.0014). However, now

Fig 4B shows that participants also selected grasps that were on average 26 mm closer to the starting location than the object centroid ($t(11)=9.74$, $p=9.6*10^{-7}$), reproducing known spatial biases in human grasp selection [15, 28, 30, 32, 33].

Consistent with Kleinholdermann et al. [15] but contrary to previous claims [18–22], our findings suggest humans care little about torque when grasping lightweight objects (of ~100 g). If actors sought to minimize torque, the selected grasps should be as close as possible to the CoM. Conversely, if participants were to disregard torque, then grasps should be at least as distant from the CoM as grasps randomly selected on the surface of the object. Fig 5A plots the difference between the CoM distance of participant grasps and the average CoM distance of random grasps, which we name 'CoM attraction compared to random grasps'. In Experiment 1, grasps were on average 9 mm farther from the CoM than the average distance to the object's CoM of grasps uniformly sampled onto the surface of the objects ($t(11)=4.53$, $p=8.6*10^{-4}$). This negative value means that participants grasped the objects towards their extremities, farther from the CoM than even random chance.

## Experiment 2: Mass and mass distribution

**Humans grasp objects close to their center of mass when high grip torques are possible and instructions demand the object does not rotate.**   Due to the low density of beech wood, even the grasps farthest from the CoM in Experiment 1 would produce relatively low torques. Therefore, in Experiment 2 we tested whether participants grasp objects closer to the CoM when higher torques are possible. We did this by using objects of greater mass and asymmetric mass distributions. Specifically, for each of the shapes in Experiment 1, we made three new objects, each made of five brass and five wooden cubes: two 'bipartite' objects, with brass clustered on one or the other half of the object, and one 'alternating' object, with brass and wood alternating along the object's length. These objects had the same 3D shapes as in Experiment 1, but were nearly tenfold heavier (Fig 2C, see Materials and methods).

Fig 5A shows how human grasps are indeed significantly attracted towards the CoM of heavy objects, presumably to counteract the larger torques associated with higher mass. In Experiment 2, grasps were on average 11 mm closer to the object CoM than grasps sampled uniformly from the objects' surfaces ($t(13)=4.89$, $p= 2.9*10^{-4}$), and on average 20 mm closer than the grasps from Experiment 1 ($t(24)=6.60$, $p= 8.0*10^{-7}$). Fig 5B shows how this behavior was evident already from the very first trial performed by participants, but also that grasps clustered more toward the object CoM in later trials, presumably as participants refined their estimates of CoM location (correlation between CoM attraction and trial repetition: $r = 0.86$, $p = 0.13$). Importantly, participants shifted their grasps towards the CoM—not the geometrical centroid—of the objects (observe how the grasp patterns shift in Fig 5C). Fig 5D shows that when the object CoM was shifted towards the hand's starting location, participants did not significantly adjust their grasping strategy compared to Experiment 1 ($t(13)=0.81$, $p=0.43$). Conversely, when the object CoM was in the same position as in Experiment 1, grasps shifted on average by 8 mm towards the CoM ($t(13)=3.92$, $p=0.0017$). When the CoM was shifted away from the hand's starting position, grasps were on average 37 mm closer to the CoM compared to Experiment 1 ($t(13)=8.49$, $p=1.2*10^{-6}$), a significantly greater shift than both the near and same CoM conditions ($t(13)=8.66$, $p=9.2*10^{-7}$ and $t(13)=7.58$, $p=4.0*10^{-6}$). These differential shifts indicate that participants explicitly estimated each object's CoM from visual material cues.

Even with the heavier objects, participants still systematically selected grasps that were closer to the starting location than the object centroid ($t(13)=4.03$, $p=0.0014$). However, now

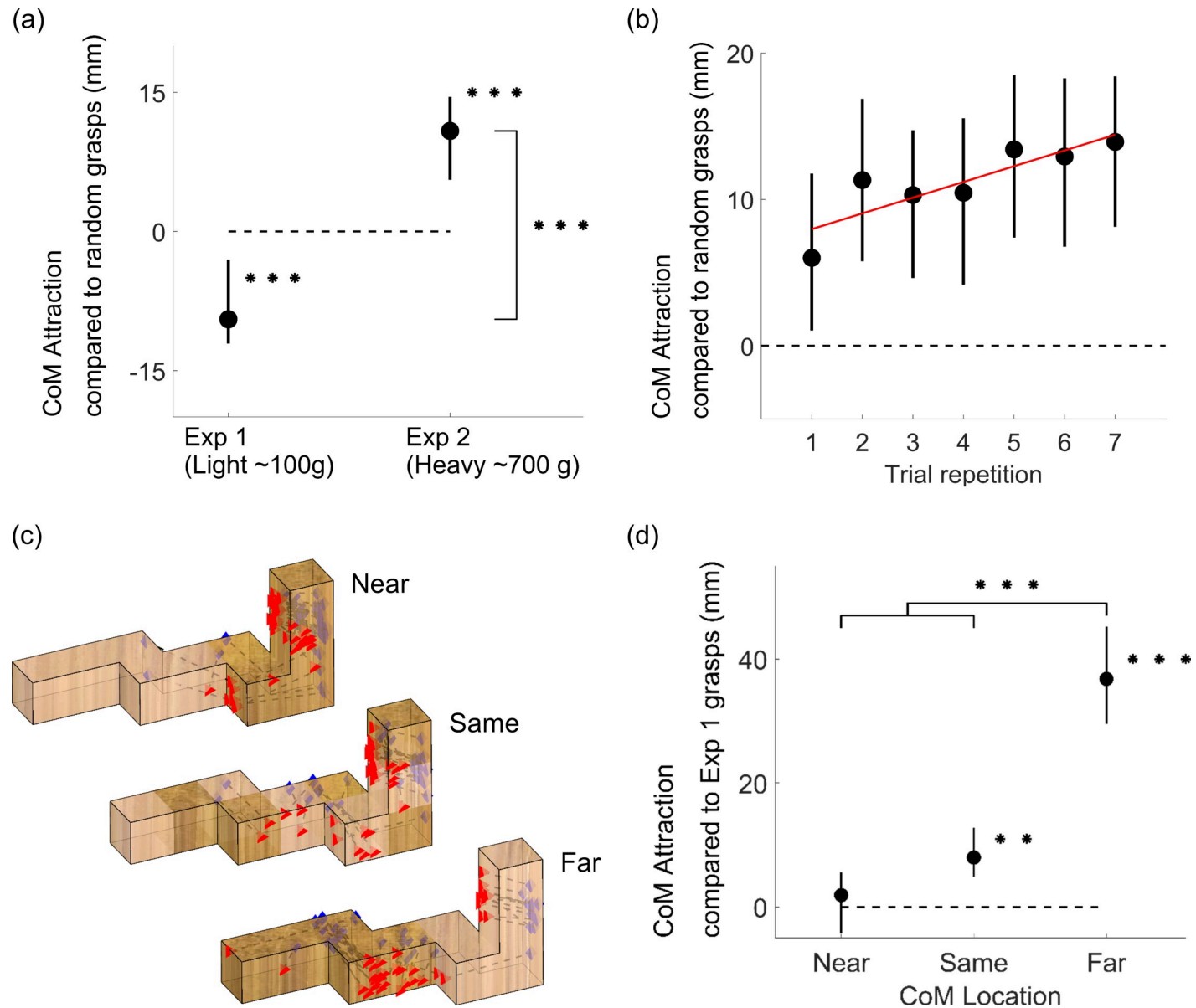

**Fig 5. Mass and mass distribution.** (a) Attraction towards the object CoM for grasps executed onto light (Experiment 1) and heavy (Experiment 2) objects compared to the average CoM distance of grasps uniformly distributed on the object surfaces (zero reference). (b) Attraction towards the object CoM in Experiment 2 as a function of trial repetition. Red line is the best-fitting regression line through the data (c) Human grasps from Experiment 2 onto object S presented at orientation 2. (d) Attraction towards the object CoM compared to Experiment 1 grasps (zero reference), for Experiment 2 grasps onto heavy objects whose CoM is closer, the same distance as, or farther than the light wooden objects from Experiment 1. In panels a, b, and d, data are means, error bars represent 95% bootstrapped confidence intervals. ** $p<0.01$, *** $p<0.001$.

participants exhibited only a 9 mm bias, which was significantly smaller than the 26 mm bias observed for the light wooden objects in Experiment 1 ($t(24)=4.67$, $p= 9.6*10^{-5}$).

Together these findings suggest that participants combine multiple constraints to select grasp locations, taking into consideration the shape, weight, orientation, and mass distribution of objects, as well as properties of their own body to decide where to grasp objects. We next sought to develop a unifying model that could predict these diverse effects based on a few simple underlying principles.

## Normative model of human grasp selection

Based on the insights gained from our empirical findings, we developed a model to predict human grasp locations. The model takes as input 3D descriptions of the objects' shape, mass distribution, orientation, and position relative to the participant, and computes as output a grasp cost function, describing the costs associated with every possible combination of finger and thumb position on accessible surface locations (i.e., those not in contact with table). We reasoned that humans would tend to grasp objects at or close to the minima of this cost function, as these would yield the most stable, comfortable grasps. Low cost grasps can then be projected back onto the object to compare against human grasps. It is important to note that this is not intended as a process model describing internal visual or motor representations (i.e., we do not suggest that the human brain explicitly evaluates grasp cost for all possible surface locations). Rather, it is a normative model for predicting which grasps are optimal under a set of pre-defined constraints. It provides a single, unifying framework based on a subset of the factors that are known to influence human grasp selection [15].

For each object, we create a triangulated mesh model in a 3D coordinate frame, from which we can sample (Fig 6A and 6B). For precision grip, we assume one contact point each for thumb and index finger. Thus, all possible precision grip grasps can be ordered on a 2D plane, with all possible thumb contact points along the x-axis, and on the y-axis, all possible index contacts in the same ordering as for the thumb.

To estimate the cost associated with each grasp, we take the combination of five penalty functions, determined by the object's physical properties (surface shape, orientation, mass, mass distribution) as well as constraints of the human actuator (i.e. the human arm/hand).

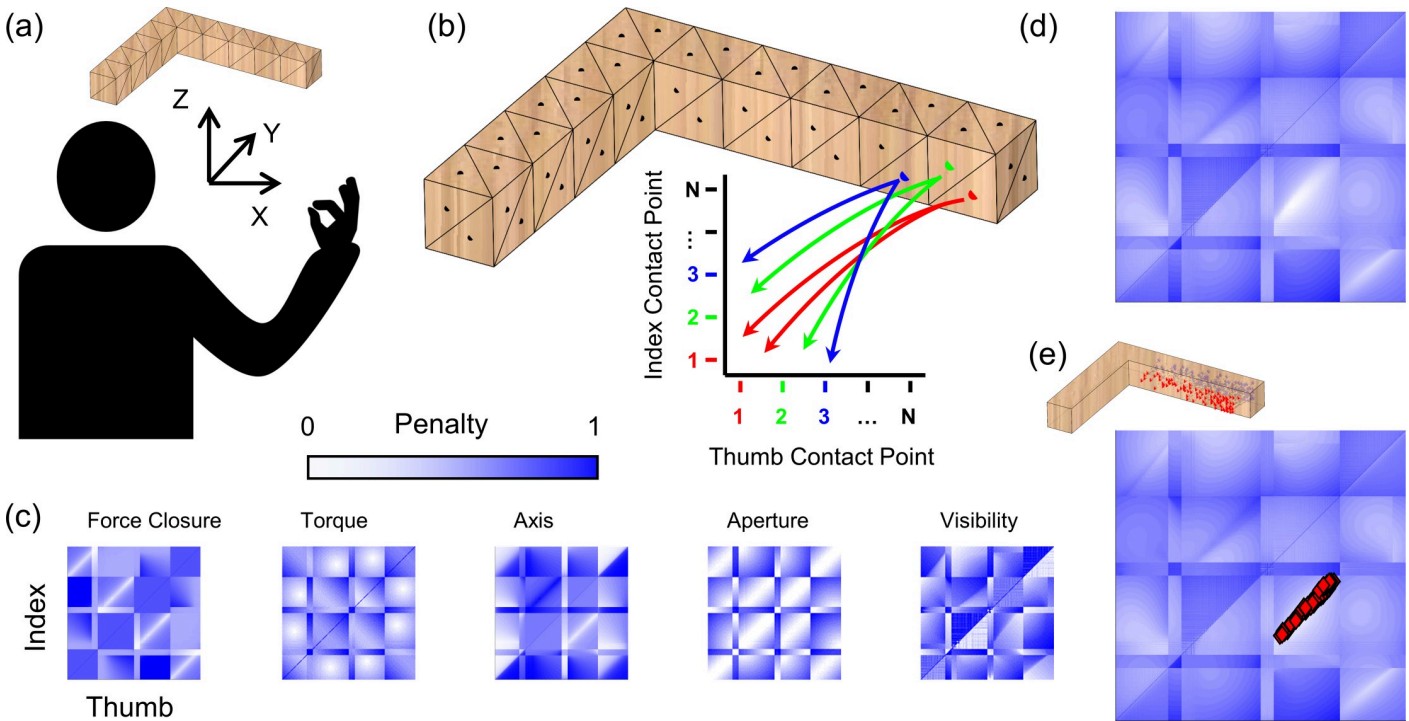

**Fig 6. A framework that unifies distinct aspects of grasp selection.** (a) Mesh model of object in same 3D reference frame as participant poised to execute grasp. (b) Discrete sampling of the reachable surface defines a 2D space containing all potential combinations of index and thumb contact points on the object. (c) Color-coded maps showing penalty values for each potential grasp for each penalty function. (d) Overall penalty function computed as the linear combination of maps in (c). (e) Human grasps projected into 2D penalty-function space neatly align with minimum of combined penalty map.

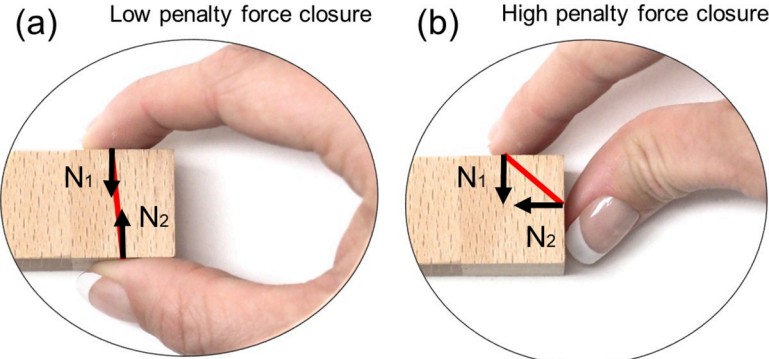

**Fig 7. Force closure.** Examples of grasps with (a) low penalty and (b) high penalty force closure.

Specifically, we consider optimality criteria based on: (i) optimum force closure [17], (ii) minimum torque [18–22], (iii) alignment with the natural grasp axis [19,23–25], (iv) optimal grasp aperture [26], and (v) optimal visibility [27,28,30] (see Materials and methods for mathematical definitions). Fig 6C shows maps for each penalty function: white indicates low penalty, dark blue high penalty. To compare and combine penalty, values are normalized to [0,1].

**Force closure:** force closure is fulfilled when the two contact-point surface normals, along which gripping forces are applied, are directed towards each other [17]. Thus, we penalize lateral offsets between the grasp point normals (Fig 7).

**Minimum torque:** grasping an object far from its CoM results in high torque, which causes the object to rotate when picked up [18–22]. Large gripping forces would be required to prevent the object from rotating. We therefore penalize torque magnitude (Fig 8).

**Natural grasp axis:** when executing precision grip grasps, humans exhibit a preferred hand posture known as the natural grasp axis [19,23–25]. Grasps that are rotated away from this axis result in uncomfortable or restrictive hand/arm configurations (Fig 9). We therefore penalize angular misalignment between each candidate grasp and the natural grasp axis (taken from [24]). Unlike force closure and torque, this penalty map is asymmetric about the diagonal: swapping index and thumb positioning produces the same force closure and torque penalties, but changes the penalty for the natural grasp axis by 180 degrees.

**Optimal grasp aperture:** for two-digit precision grips humans prefer the distance between finger and thumb at contact ('grasp aperture') to be below 2.5 cm [26]. We therefore penalize grasp apertures above 2.5 cm (Fig 10).

**Optimal visibility:** our behavioral data, and previous studies, suggest humans exhibit spatial biases when grasping. It has been proposed that these may arise from an attempt to

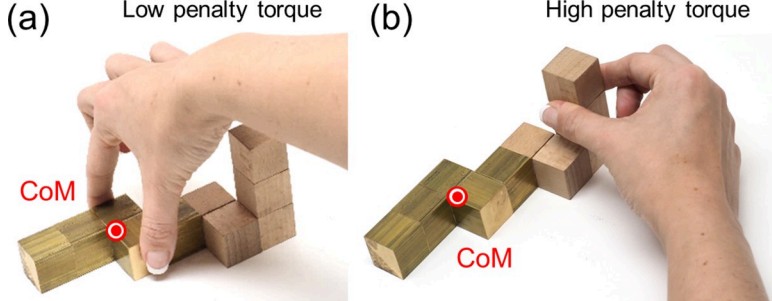

**Fig 8. Torque.** Examples of grasps with (a) low penalty and (b) high penalty torque.

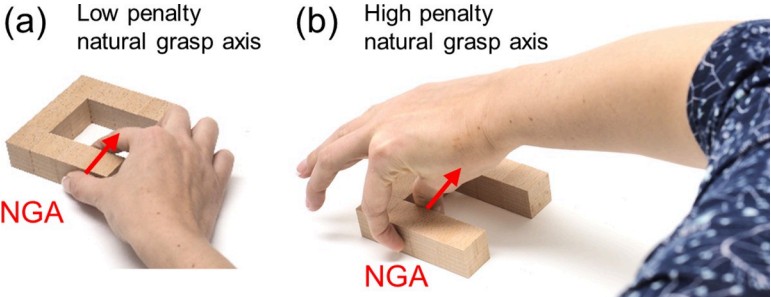

**Fig 9. Natural grasp axis.** Examples of grasps with (a) low penalty and (b) high penalty grasp axis.

minimize energy expenditures through shorter reach movements [27]. However, Paulun et al. [28] have shown that these biases may in fact arise from participants attempting to optimize object visibility. While our current dataset was not designed to untangle these competing hypotheses, re-analyzing published data [22,30] confirms that object visibility—not reach length—is most likely responsible for the biases. We therefore penalized grasps that hindered object visibility (Fig 11). We also designed a penalty function for reach length and verified that, since reach length and object visibility are correlated in our dataset, employing one or the other penalty function yields very similar results.

   We assume that participants select grasps with low overall costs across all penalty functions. Thus, to create the overall grasp penalty function, we take the sum of the individual penalty maps. The minima of this full penalty map represent grasps that best satisfy all criteria simultaneously. The map in Fig 6D exhibits a clear minimum: the white region in its lower right quadrant.

   To assess the agreement between human and optimal grasps, we may visualize human grasps in the 2D representation of the grasp manifold. The red markers in Fig 6E are the human grasps from object L at orientation 2, projected in 2D and overlain onto the full penalty map. Human grasps neatly align with the minima of the penalty map, suggesting that human grasps are nearly optimal in terms of the cost criteria we use.

   **Model fitting.**   The simple, equal combination of constraints considered thus far already agrees with human grasping behavior quite well. However, it is unlikely that actors treat all optimality criteria as equally important. Different persons likely weight the constraints differently (e.g., due to strength or hand size). Therefore, we developed a method for fitting full penalty maps to participants' responses. We assigned variable weights to each optimality criterion, and fit these weights to the grasping data from each participant, to obtain a set of full penalty maps whose minima best align with each participant's grasps (see Materials and methods).

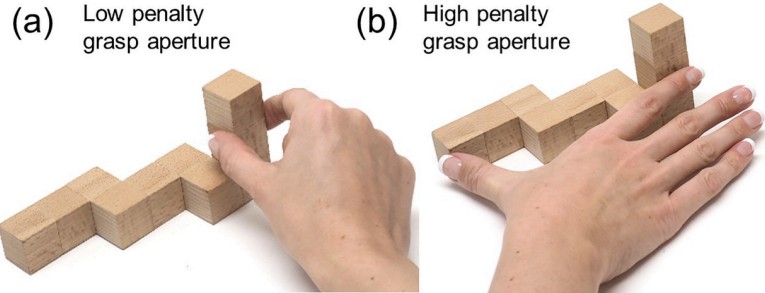

**Fig 10. Optimal grasp aperture.** Examples of grasps with (a) low penalty and (b) high penalty aperture.

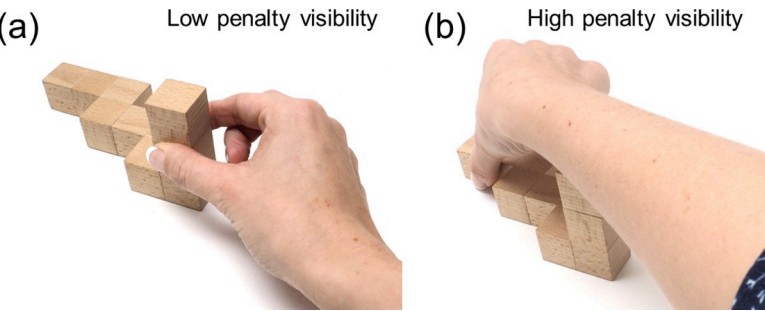

**Fig 11. Optimal visibility.** Examples of grasps with (a) low penalty and (b) high penalty visibility.

**Model grasps are nearly indistinguishable from measured human grasps.**  To compare human and optimal grasps directly, we can sample predicted optimal grasps from around the minimum of the full penalty map (see Materials and methods) and project back onto the objects. Fig 12A shows human grasps (left) and unfitted model predictions (right) on a few representative objects (see S1 Fig for complete set). Human and predicted grasps have similar size and orientation, and also cover similar portions of the objects.

Fig 12B depicts grasp similarity at the population level, i.e., across participants and between human and unfitted model grasps. Grasp similarity between participants was computed (for each object and condition), as the similarity between the medoid grasp of each participant and the medoid grasp across all others. Grasp similarity between human and model grasps was computed as the similarity between the medoid unfitted model grasp and the medoid grasp across all participants.

Unfitted model grasps were significantly more similar to human grasps than chance (t(31) =9.34, p=$1.6*10^{-10}$), and effectively indistinguishable from human-level grasps similarity (t(31) =0.53, p=0.60). Note that this does not mean our current approach perfectly describes human grasping patterns; it suggests instead that our framework is able to predict the medoid human grasping patterns nearly as well as the grasps of a random human on average approximate the medoid human grasp.

**Fitting the model can account for individual grasp patterns.**  In both Experiments, participants repeatedly grasped the same objects in randomized order. Fig 12C depicts how similar human and model grasps are to the medoid grasp of each individual participant in each experimental condition. Individual subjects are highly consistent when grasping the same object on separate trials. Grasps predicted through our framework with no knowledge of the empirical data were significantly less similar to the medoid grasps of individual humans (t(31)=9.28, p=$1.9*10^{-10}$). This is unsurprising, since the unfitted model predicts the average pattern across observers, but there is no mechanism for it to capture idiosyncrasies of individual humans. Fitting the model to the human data (see Materials and methods) significantly improved grasp similarity (t(31) =4.26, p=$1.8*10^{-4}$). Note however that model grasp patterns fit to a single participant are still distinguishable from random real grasps by the same individual (t(31)=4.91, p=$2.8*10^{-5}$).

**Force closure, hand posture, and grasp size explain most of human grasp point selection.**  The pattern of fitted weights across both experiments (Fig 12D) reveals the relative importance of the different constraints. Specifically, we find that force closure is the most important constraint on human grasping, which makes sense because force closure is a physical requirement for a stable grasp. Next in importance are natural grasp axis and optimal grasp aperture, both constraints given by the posture and size of our actuator (our hand). In comparison, participants appear to care only marginally about minimizing torque, and almost negligibly about object visibility.

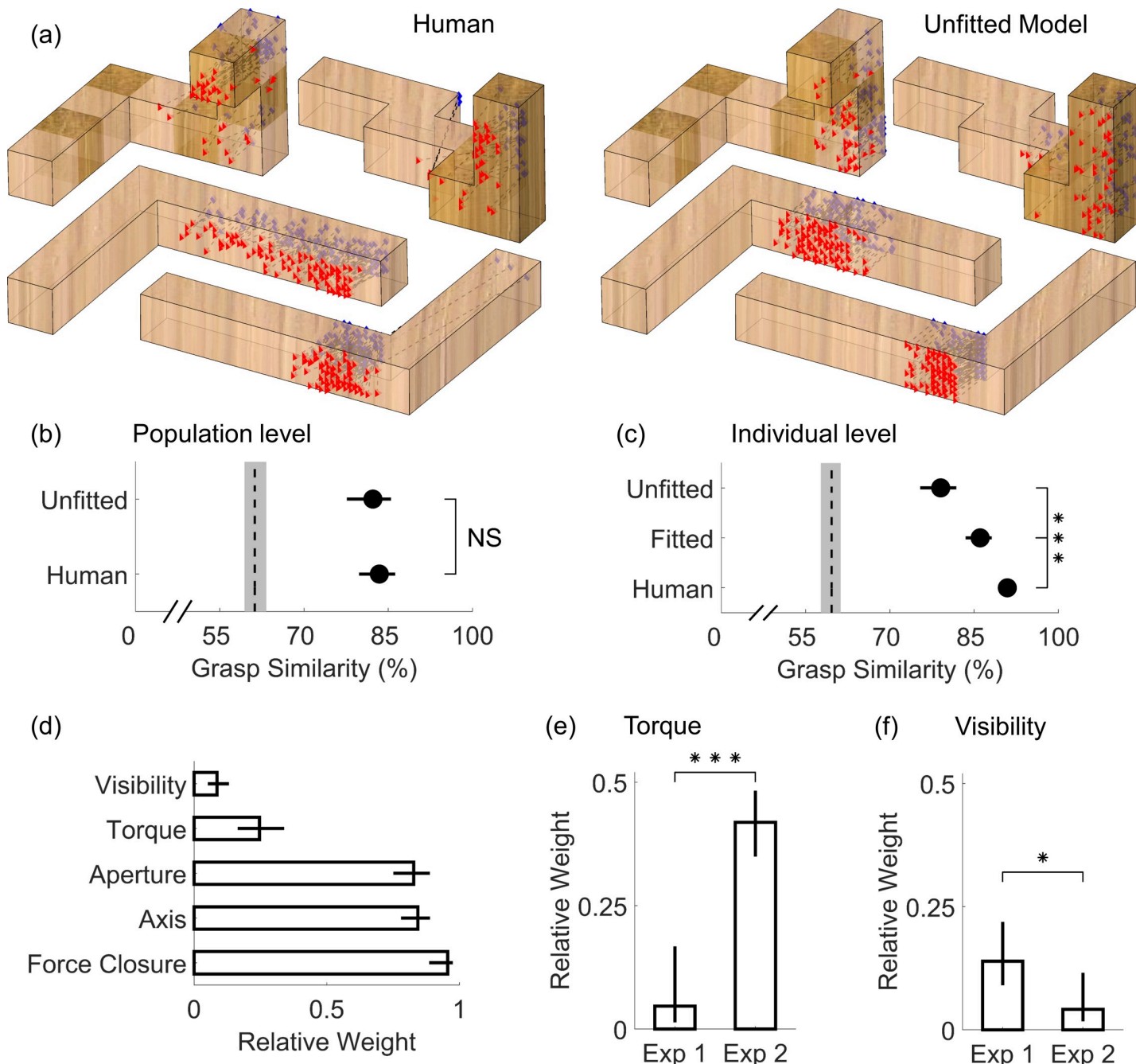

**Fig 12. Model results.** (a) Grasping patterns reconstructed through the normative framework (right) closely resemble human grasps onto real objects varying in shape, orientation, and material (left). Simulated grasp patterns are generated with no knowledge of our human data (i.e. model not fit to human grasps). (b) Population level grasp similarity, i.e. similarity of human and unfitted model grasps to medoid human grasp across all participants. (c) Individual level grasp similarity, i.e. similarity of human, unfitted, and fitted model grasps to the medoid grasp of each participant. In panels (b, c), dashed line is estimated chance level of grasp similarity due to object geometry, bounded by 95% bootstrapped confidence intervals. (d) Pattern of fitted weights across Experiments 1 and 2. (e) Relative weight of the minimum torque constraint in Experiments 1 and 2. (f) Relative weight of the visibility constraint in Experiments 1 and 2. Data are means; error bars, 95% bootstrapped confidence intervals. ***p<0.001.

**Analyzing the patterns of fitted weights confirms our empirical findings.** The model also replicates our main empirical findings in a single step. Fig 12E shows that the relative importance of torque was much greater for the heavy objects tested in Experiment 2 compared

to the light objects from Experiment 1 (t(24)=7.93, p=3.7*10^{-8}). Conversely, Fig 12F shows that the relative importance of object visibility instead decreased significantly from Experiment 1 to Experiment 2 (t(24)=2.62, p=0.015). Additionally, by simulating grasps from the fitted model, we are able to recreate the qualitative patterns of all behavioral results presented in Figs 3,4 and 5 (see S2 Fig).

### Experiment 3: Model validation

To further validate the model, we tested whether the model makes sensible predictions on novel objects and whether the model is robust to perturbations.

**Model predictions on novel objects.**   The model was designed from the insights derived from Experiments 1 and 2 with polycube objects made of brass and wood. To test whether the model generalizes beyond this type of object, we selected four mesh models of objects with smooth, curved surfaces from an in-house database (two familiar, two unfamiliar objects). We input these meshes to the model and generated grasp predictions (Fig 13A). The model was instantiated using the weights derived from Experiment 1. Next, we 3D printed these objects out of light plastic (~80g, comparable to Experiment 1 objects), and asked 14 human participants to grasp these novel objects. Fig 13B shows how human grasps agree with model predictions. Human and model grasps once again have similar size and orientation, and also cover similar portions of the objects. Fig 13C confirms this observation: predicted model grasps are as similar to medoid human grasps as grasps from a random human participant (t(13)=1.21, p=0.25).

**Model perturbation analysis.**   The model designed thus far receives as input a near-veridical representation of the objects to grasp. However, it is unlikely that humans have access to such a veridical object representation. We therefore implemented some perturbations to the inputs and key parameters of the model and observed how robust the model is to these perturbations. Specifically, we tested how model performance in predicting human grasping patterns from Experiment 3 varies as a functions of these perturbations.

The model input thus far consisted of densely sampled 3D mesh models. It's unlikely that humans also have such a dense, accurate 3D representation of an object's surface. Fig 14A therefore shows model performance (in terms of similarity with human grasping patterns) with different levels of surface mesh subsampling. Model performance is robust to relatively high levels of subsampling, and decreases only once sampled surface locations are on average more than 4 mm distant from one another (below 5% mesh subsampling).

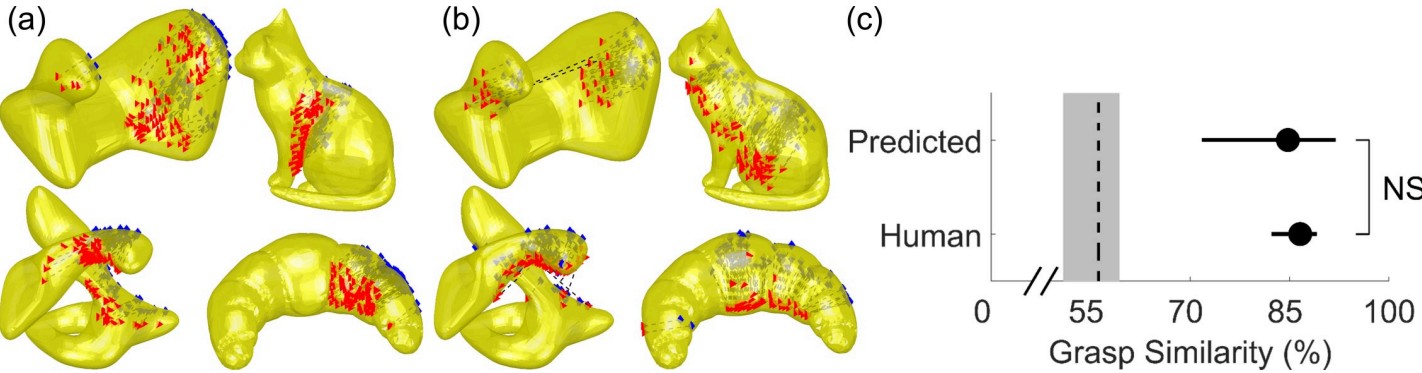

**Fig 13. Model predictions for novel objects align with human grasps.** (a) Grasping patterns predicted through the normative framework for novel objects with smooth and curved surface geometry. (b) Human grasps onto 3D printed versions of the objects align with model predictions. (c) Similarity of human and predicted model grasps to medoid human grasp across objects and participants. Dashed line is estimated chance level of grasp similarity, bounded by 95% bootstrapped confidence intervals.

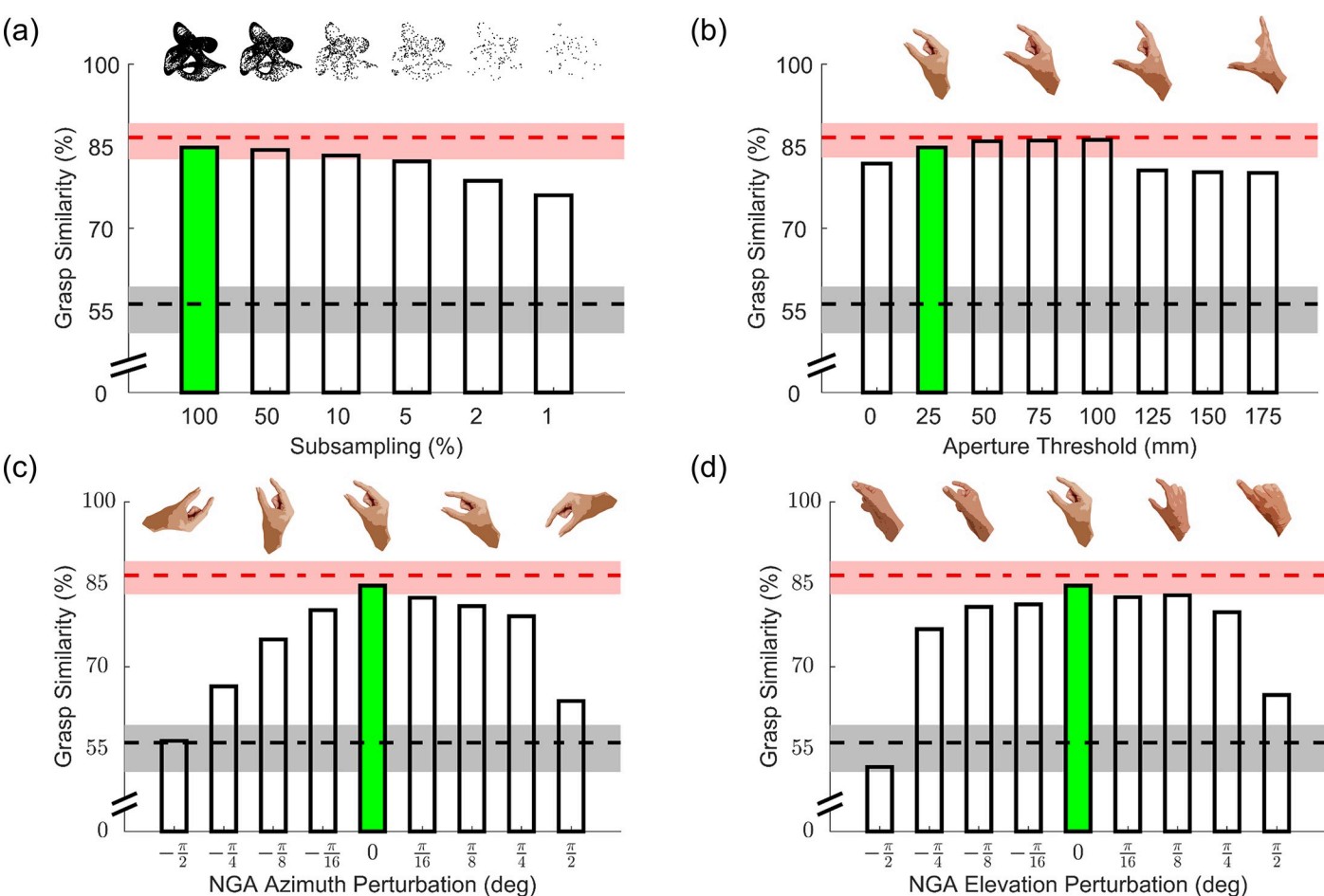

**Fig 14. Perturbation results.** All panels show model performance (in terms of grasp similarity to human data from Experiment 3) as a function of different perturbations. Grasp similarity for the original model implementation is shown in green. Red and black dashed lines are respectively human and chance levels of grasp similarity, bounded by 95% bootstrapped confidence intervals. (a) Model grasp similarity with input meshes subsampled by varying degrees. (b) Model grasp similarity for model implementations employing increasing aperture thresholds. (c, d) Model grasp similarity for models implemented with deviated natural grasp axis along the transverse (c) and sagittal (s) planes.

Since the backside of objects is occluded from view, it is unlikely that participants have an accurate estimate of the required grip aperture across the whole object. Additionally, since we constrained participants to two-digit precision grips, grasps above the threshold defined by Cesari and Newell [26] might be acceptable, as long as these are within a maximum comfortable grasp span. Fig 14B shows that indeed model performance is robust to increases in aperture threshold up to 100 mm.

Similarly, humans might also exhibit some tolerance for grasps oriented away from the natural grasp axis. Given that the ease of a rotation of the arm and hand is likely asymmetric along different directions, these tolerances likely also vary depending on rotation direction. Fig 14C shows how model performance does indeed decrease for perturbations of the natural grip axis along the transverse plane, and this decrease is more steep for clockwise (negative) rotations, as already suggested by Kleinholdermann and colleagues [15]. Model performance is instead more robust to perturbations along the sagittal plane (Fig 14D), and particularly for (positive) counterclockwise rotations in which the thumb tilts below the index finger.

## Discussion

We investigated how an object's 3D shape, orientation, mass, and mass distribution jointly influence how humans select grasps. Our empirical analyses showed that grasping patterns are highly systematic, both within and across participants, suggesting that a common set of rules governs human grasp selection of complex, novel 3D objects. Our findings reproduce, unify, and generalize many effects observed previously: (i) both 3D shape and orientation determine which portion of the object people grasp [8,15,18,19,34–37]; (ii) humans exhibit spatial biases even with complex 3D objects varying in shape and mass [15,28,30,32,33]; (iii) object weight modulates how much humans take torque into account when selecting where to grasp objects [18–22]. We then combined this diverse set of observations into a unified theoretical framework that predicts human grasping patterns strikingly well, even with no free parameters. By fitting this normative model to human behavioral data, we showed that force closure, hand posture, and grasp size are the primary determinants of human grasp selection, whereas torque and visibility modulate grasping behavior to a much lesser extent. We further demonstrated that the model is able to generate sensible predictions for novel objects and is robust to perturbations.

### 3D shape

Behavioral research on the influence of shape on grasping is surprisingly scarce, primarily employs 2D or simple geometric 3D stimuli of uniform materials, and rarely investigates grasp selection [8, 18, 19, 34–37]. For example, by using 3D stimuli that only varied in shape by a few centimeters, Schettino et al. [36] concluded that object shape influences hand configuration only during later phases of a reaching movement during which subjects use visual feedback to optimize their grasp. Here, we show that distinct 3D shapes are grasped in systematically distinct object locations, and our behavioral and model analyses can predict these locations directly from the object 3D shape.

### Orientation

When grasping spheres or simple geometrical shapes, humans exhibit a preferred grasp orientation (the NGA) [19,23–25], and most previous work on how object orientation influences grasping has primarily focused on hand kinematics [18,22,35,38]. Conversely, with more complex 3D shapes we show that the same portion of an object is selected within a range of orientations relative to the observer, whereas for more extreme rotations the grasp selection strategy shifts significantly. Therefore, object shape and orientation together determine which portion of an object will be grasped, and thus the final hand configuration.

### Spatial biases

The spatial biases we observe are consistent with participants attempting to increase object visibility [28,30], and our data also replicate the finding that these biases are reduced when object weight increases [22,28].

### Material/weight/torque

Goodale et al. [18] were among the first to show that participants tend to grasp objects through their CoM, presumably to minimize torque. Lederman and Wing [19] found similar results, yet in both studies low-torque grasps also correlated with grasps that satisfied force closure and aligned with the natural grasp axis. Kleinholdermann et al. [15] found torque to be nearly irrelevant in grasp selection, yet Paulun et al. [22] observed that grasp distance to CoM was

modulated by object weight and material. More recent work by Paulun et al. has further shown that participants are fairly accurate at visually judging the location of the CoM even for bipartite objects made of two different materials [39]. Our findings resolve these conflicting findings. By using stimuli that decorrelate different aspects of grasp planning, we find that shape and hand configuration are considerably more important than torque for light weight objects, and that the importance of minimizing torque scales with mass. Additionally, shifting an object's mass distribution significantly attracted grasp locations towards the object's shifted CoM, demonstrating that participants could reliably combine global object shape and material composition to successfully infer the object's CoM.

## Modelling grasp selection

Previous models of grasping have mainly focused on hand kinematics and trajectory synthesis [2–6] whereas we attempt to predict which object locations will be selected during grasping. Our modelling approach takes inspiration from Kleinholdermann et al. [15], which to the best of our knowledge is the only previous model of human two-digit contact point selection, but only for 2D shape silhouettes. In addition to dealing with 3D objects varying in mass, mass distribution, orientation, and position, our modeling addresses several limitations of previous approaches. The fitting procedure quantifies the relative importance of different constraints, and can be applied to any set of novel objects to test how experimental manipulations affect this relative weighting. Additionally, while model fitting significantly improved the similarity between model and individual participant grasps, the agreement was not perfect. This suggests that grasp planning may involve additional, undiscovered constraints, which our approach would be sensitive enough to detect. The modular nature of the model specifically allows additional constraints to be included, excluded or given variable importance. For example, we know that end-state comfort of the hand plays a role in grip selection [40,41], yet the tradeoff between initial and final comfort is unclear [42]. By varying the participants' task to include object rotations, and by including a penalty function penalizing final hand rotations away from the natural grasp axis, it would be possible to assess the relative importance of initial, final (or indeed intermediate) hand configurations on grasp planning. Relatedly, the effect of obstacles (and self-obstacles, such as the vertically protruding portions of some of the objects employed in this study) could also be assessed. The presence of obstacles could affect grasp selection by requiring reach-to-grasp trajectories that avoid an obstacle, although previous research has shown that forcing different hand paths does not affect selected grasp locations [25]. Alternatively, the presence of obstacles might alter the configuration of the arm and hand during a grasp [43], which could be incorporated into the model by modifying the grip comfort penalties.

Previous literature has also shown that object surface properties such as curvature [13], tilt [14], and friction [44,45] modulate the fingertip forces employed during grasping. While the current study was not designed to examine how these factors influence grasp selection, the current model is already able to predict grasp patterns for objects with curved surfaces, even if not perfectly. Model performance with these objects could likely be improved by including into our framework penalty functions that take into account local surface structure and friction. Incorporating friction into the model could even improve model performance for our composite objects from Experiment 2, as wood and brass may have different friction coefficients. Since surface friction plays a decisive role in determining force closure, friction coefficients could even be directly integrated into the force closure computations. Friction is also a particularly interesting test case for our assumption of a weighted linear combination of costs, as it may interact with other factors. When friction is low, it could cause the cost of torque to be

upregulated, to avoid slipping [22]. This would require the addition of parameters describing interactions between factors. Alternatively, friction and torque might be unified into a single penalty function capturing the magnitude of grip force required to avoid slippage. However, incorporating friction into the model would be non-trivial, since the coefficient of friction between skin and different materials depends on several factors, including temperature, hydration, and age [46].

The model should also be extended to multi-digit grasping, by adding to each penalty function three dimensions for each additional finger considered (the x,y,z coordinates of the contact point). This approach is consistent with (and complementary to) the approach by Smeets and Brenner [2,5], who posit that grasping is a combination of multiple pointing movements. Given that human participants adjust the number of digits they employ to grasp an object depending on grip size and object weight [26], multiple size/weight thresholds could be employed to determine the preferred multi-digit grip. Future models should also generalize from contact points to contact patches of nonzero area, as real human grasp locations are not only points but larger areas of contact between digit and object. To facilitate such developments, we provide all data and code (doi: 10.5281/zenodo.3891663).

### Neuroscience of grasping

While our model is not intended as a model of brain processes, there are several parallels with known neural circuitry underlying visual grasp selection (for reviews see [47–49]). Of particular relevance is the circuit formed between the Ventral Premotor Cortex (Area F5), Dorsal Premotor Cortex (Area F2), and the Anterior Intraparietal Sulcus (AIP). Area F5 exhibits 3D-shape-selectivity during grasping tasks and is thought to encode grip configuration given object shape [50–52], whereas area F2 encodes the grip-wrist orientation required to grasp objects under visual guidance [53]. Both regions exhibit strong connections with AIP, which has been shown to represent the shape, size, and orientation of 3D objects, as well as the shape of the handgrip, grip size, and hand-orientation [54]. Additionally, visual material properties, including object weight, are thought to be encoded in the ventral visual cortex [55–59], and it has been suggested that AIP might play a unique role in linking components of the ventral visual stream involved in object recognition to hand motor system [60]. Therefore, the neural circuit formed between F5, F2, and particularly AIP is a strong candidate for combining the multifaceted components of visually guided grasping identified in this work [61–65]. Combining targeted investigations of brain activity with the behavioral and modelling framework presented here holds the potential to develop a unified theory of visually guided grasp selection.

## Materials and methods

### Ethics statement

All procedures were approved by the local ethics committee of the Department of Psychology and Sports Sciences of the Justus-Liebig University Giessen (Lokale Ethik-Kommission des Fachbereichs 06, LEK-FB06; application number: 2018-0003) and adhered to the declaration of Helsinki. All participants provided written informed consent prior to participating.

### Participants

Twelve naïve participants (5 males and 7 females between the ages of 20 and 31, mean age: 25.2 years) participated in Experiment 1. A different set of fourteen naïve participants (9 males and 5 females between the ages of 21 and 30, mean age: 24.4 years) participated in Experiment 2. An additional, different set of fourteen naïve participants (5 males and 9 females between

the ages of 19 and 58, mean age: 25.1 years) participated in Experiment 3. Participants were students at the Justus Liebig University Giessen, Germany and received monetary compensation for participating. All participants reported having normal or corrected to normal vision and being right handed.

## Apparatus

Experiments 1 and 2 were programmed in Matlab version R2007a using the Optotrak Toolbox by V. H. Franz [66]. Participants were seated at a table with their head positioned in a chinrest (Fig 2A), in front of an electronically controlled pane of liquid crystal shutter glass [67], through which only part of the table was visible and which became transparent only for the duration of a trial. Objects were placed at a target location, 34 cm from the chinrest in the participant's sagittal plane. Small plastic knobs placed on participants' right side specified the hand starting positions. A plate (28.5 cm to the right of the target location and with a 13 cm diameter at 26 cm from start position 1 in the participant's sagittal plane) specified the movement goal location. We tracked participants' fingertip movements with sub-millimeter accuracy and resolution using an Optotrak 3020 infrared tracking system. The Optotrak cameras were located to the left of the participants. To record index finger and thumb movement, sets of three infrared markers (forming a rigid body) were attached to the base of the participants' nails. The fingertip and tip of the thumb were calibrated in relation to the marker position, as participants grasped a wooden bar with a precision grip, placing their fingertips at two known locations on the bar.

Experiment 3 was programmed in Matlab version R2019b using the Motom Toolbox [68]. Participants were seated at a table with their head positioned in a chinrest and had their eyes open only for the duration of the movement execution (Fig 15A). Objects were placed at a target location, 36 cm from the chinrest in the participant's sagittal plane. A piece of tape placed 30 cm to the right of the chinrest specified the hand starting position. A plate (30 cm to the right of the target location and with an 18 cm diameter at 30 cm from the start position in the participant's sagittal plane) specified the movement goal location. We tracked participants' fingertip movements using an Optotrak Certus infrared tracking system. The Optotrak cameras were located to the left of the participants. To record index finger and thumb movement, sets of three infrared markers (forming a rigid body) were attached to the base of the participants' nails. The fingertip and tip of the thumb were calibrated in relation to the marker position, as participants touched another marker using a precision grip, placing their finger- and thumb tip at the center of the marker one after the other.

## Stimuli

**Experiment 1: Light objects made of wood.**   Four differently shaped objects (defined as objects L, U, S and V; Fig 2B) each composed of 10 wooden (beech) cubes ($2.5^3$ cm$^3$), served as stimuli. Objects were fairly light with a mass of 97 g. Two of the objects featured cubes stacked on top of each other, whereas the other two objects were composed exclusively of cubes lying flat on the ground. The objects were presented to the participants at one of two orientations. Across orientations, object L was rotated by 180 degrees, objects U and V were rotated by 90 degrees, and object S was rotated by 55 degrees. Fig 2B shows the objects positioned as if viewed by a participant.

**Experiment 2: Heavy composite objects made of wood and brass.**   For each of the 4 shapes from Experiment 1, we created 3 new objects (12 in total) to serve as stimuli for Experiment 2 (Fig 2C). Individual cubes were made of either wood or brass. The objects were composed of 5 cubes of each material, which made them fairly heavy with a mass of 716g. By reordering the sequence of wood and brass cubes, we shifted the location of each shape's CoM.

For each shape we made one object in which brass and wooden cubes alternated with one another, and two bipartite objects, where the 5 brass cubes were connected to one another to make up one side of the object with the wooden cubes making up the other side. This configuration was also inverted, (i.e., wooden and brass cubes switched locations). The 'alternating' objects had approximately the same CoM as their wooden counterparts (mean ± sd distance: 5.1±2.5 mm). Conversely, the CoM of bipartite objects was noticeably shifted to one side of the object compared to their wooden counterparts (mean ± sd distance: 33.3±4.4 mm). The CoM locations for all stimuli are shown in S3 Fig. All objects were presented at the same two orientations as Experiment 1.

**Experiment 3: Curved 3D-printed object.**   Four novel, differently shaped objects were 3D-printed. They were made from a yellow plastic with a stabilizing mesh inside. Two objects were abstract, curved shapes objects (defined as 'swan' (64g) and 'blob' (121g), the other two objects were known shapes: a cat (72g) and a croissant (74g). All objects were presented to participants in one orientation, as displayed in Fig 15B.

**Object meshes.**   For Experiments 1 and 2 triangulated mesh replicas of all objects were created in Matlab; each cube face consisted of 128 triangles. For Experiment 3 we selected non-uniform mesh model objects from an in-house database, each mesh consisting of between 4500 and 9000 triangles. To calibrate mesh orientation and position, we measured, using the Optotrak, four non planar points on each object at each orientation. We aligned the model to the same coordinate frame employed by the Optotrak using Procrustes analysis.

## Procedure

**Experiments 1 and 2.**   Prior to each trial, participants placed thumb and index finger at a pre-specified starting location. In Experiment 1, two start locations were used (start 1 at 28 cm

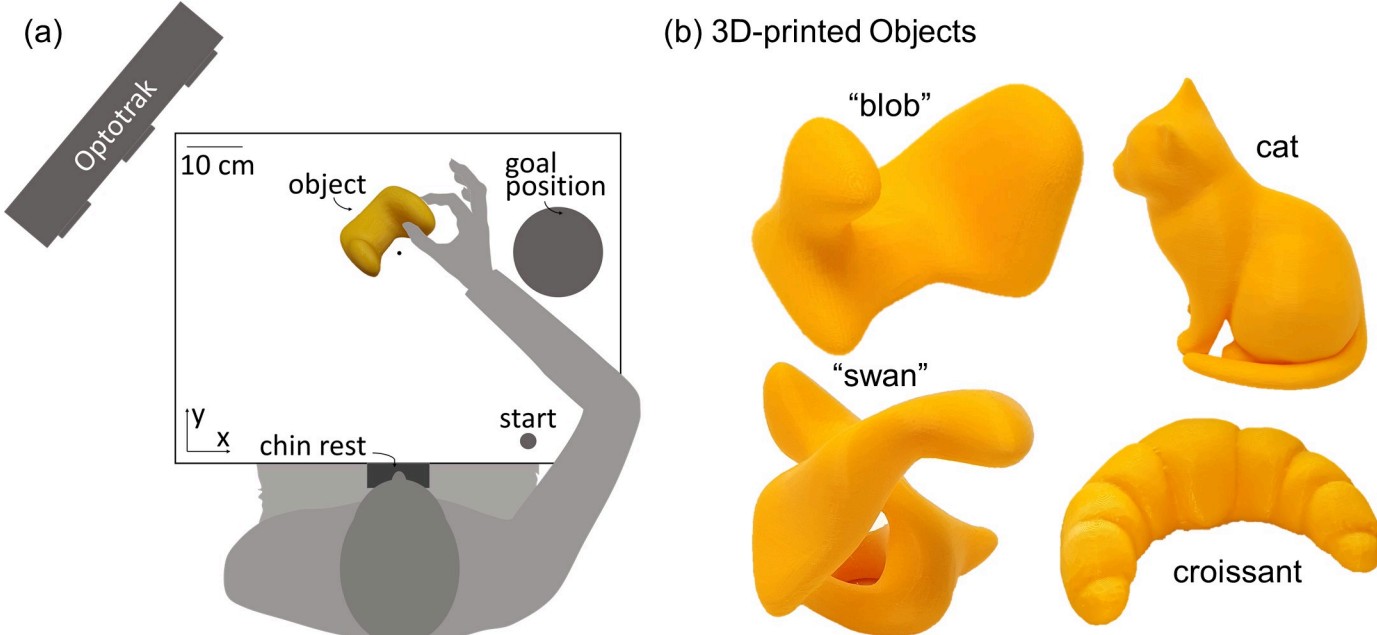

**Fig 15. Setup and stimuli for Experiment 3.** (a) Experimental setup. Seated participants performed grasping movements with their right hand. Following an auditory signal, they opened their eyes, and moved from the starting position to the object and grasped it with a precision grip. They transported and released the object at the goal position and returned to the start position. (b) We employed four 3D-printed objects. Two objects had an abstract shape (that here we name 'swan' and 'blob'), the other two objects were printed versions of a croissant and a cat. They were presented to the participant in the orientations displayed in here.

to the right of the chinrest in the participant's coronal plane and 9.5 cm forward in the sagittal plane; start 2 9 cm further to the right and 3 cm further forward, 23 cm from the center of the goal plate). Given that we observed no effect of starting position in our data, in Experiment 2 only the first starting location was employed. When the subject was at the correct start position, the experimenter placed one of the stimulus objects at the target location behind the opaque shutter screen. Each object could be presented at one of two orientations with respect to the participant. The experimenter could very precisely position each object at the correct location and orientation by aligning two small groves under each object with two small pins on the table surface.

Once both stimulus and participant were positioned correctly, a tone indicated the beginning of a trial, at which point the shutter window turned translucent. Participants were then required to pick up the object using only forefinger and thumb and place it at the goal location. Participants had 3 seconds to complete the task before the shutter window turned opaque. In Experiment 1, no instructions were given regarding how the objects had to be transported, yet we observed that participants never allowed the objects to rotate. Therefore, to match the movement task across experiments, in Experiment 2 participants were instructed to keep the objects as level as possible.

Experiment 1 had sixteen conditions: two starting locations, four wooden objects of different shapes, each object presented at two orientations. Each participant repeated each condition five times (eighty trials per participant).

Experiment 2 had thirty-six conditions: twelve distinct objects (four shapes in three material configurations) presented at two orientations. Half of the participants handled only shapes L and V, the other half handled shapes U and S. Each participant repeated each condition seven times (eighty-four trials per participant). In both experiments trial order was randomized.

Following each trial, the experimenter visually inspected the movement traces to determine whether the trial was successful or not. Unsuccessful grasps were marked as error trials, added to the randomization queue, and repeated.

**Experiment 3.** Prior to each trial, participants placed thumb and index finger at the starting location, closed their eyes, and the experimenter placed one of the stimulus objects at the target location. The experimenter could precisely position each object by aligning it with its outline, drawn on millimeter paper. Once both stimulus and participant were positioned correctly, a tone indicated the beginning of a trial, at which point the participants opened their eyes. Participants were then required to pick up the object using only forefinger and thumb and place it at the goal location. Participants had 3 seconds to complete the task. Each participant picked up each object seven times (28 trials per participant). Trial order was randomized. Following each trial, the experimenter visually inspected the movement traces to determine whether the trial was successful or not. Unsuccessful grasps were marked as error trials, and repeated immediately.

**Error trials.** A total of 397 error trials (13.8% of trials from Experiment 1, 13.9% from Experiment 2, and 6.9% from Experiment 3) were not analyzed. Trials were deemed unsuccessful when participants did not conclude the movement within the allotted time (10.1% of error trials in Experiment 1, 41.4% of error trials in Experiment 2, and 0% in Experiment 3), and/or when tracking was lost (94.2% of error trials in Experiment 1, 88.7% of error trials in Experiment 2, and 100% of error trials in Experiment 3), or when participants placed the objects too hastily on the goal location, which resulted in the objects toppling over off the goal plate where they were supposed to rest (this occurred only twice throughout the study). Note that there was some overlap between causes of error. The trajectories of lost-tracking error trials, where the data are available, fall within the clusters of trajectories of corresponding non-

error trials in 92.2% and 99.0% of cases across Experiments 1 and 2 respectively. In Experiment 3 the experimenter manually recorded grasp locations for error trials, and these locations are all represented in the final dataset. It is therefore unlikely that excluded error trials differed strongly from the data included in our analyses.

## Training

At the beginning of the experiments, each participant completed six practice trials in Experiments 1 and 2 (using a Styrofoam cylinder in Experiment 1, and by lifting random objects from the shapes not used in that participant's run in Experiment 2) and five practice trials in Experiment 3 (using the wooden L-object from Experiment 1). This was done to give participants a sense for how fast their movement should be in order to complete the entire movement within three seconds. Prior to Experiment 2, participants were familiarized with the relative weight of brass and wood using two rectangular cuboids of dimensions 12.5x2.5x2.5 cm, one of wood (50 g) and one of brass (670 g). Practice trial data were not used in analyses. Prior to Experiment 3, participants were familiarized with the weight of all four test objects by having each object placed on the flat, extended palm of their right hand.

## Analyses

All analyses were performed in Matlab version R2018a. Differences between group means were assessed via paired or unpaired t-tests, or through Pearson correlation, as appropriate. Values of $p < 0.05$ were considered statistically significant.

**Contact points.** Contact points of both fingers with the object were determined as the fingertip coordinates at the time of first contact, projected onto the surface of the triangulated mesh models of the object. The time of contact with the object was determined using the methods developed by Schot et al. [69] and previously described in Paulun et al. [22].

**Grasp similarity.** We described each individual grasp $\overrightarrow{G}$ as a 6D vector of the x-, y-, z-coordinates of the thumb and index finger contact points:

$$\overrightarrow{G} = [x_T, y_T, z_T, x_I, y_I, z_I]$$

To compute the similarity $S$ between two grasps $\overrightarrow{G_1}$ and $\overrightarrow{G_2}$, we first computed the Euclidian distance between the two 6D grasp vectors. We then divided this distance by the largest possible distance between two points on the specific object $D_{max}$, determined from the mesh models of the objects. Finally, similarity was defined as 1 minus the normalized grasp distance, times 100:

$$S = 100 * \left( 1 - \frac{\| \overrightarrow{G_1} - \overrightarrow{G_2}, \|}{D_{max}} \right)$$

In this formulation, two identical grasps, which occupy the same point in a 6D space, will be 100% similar, whereas the two farthest possible grasps onto a specific object will be 0% similar. Within-subject grasp similarity was the similarity between grasps from the same participant to the participant's own medoid grasp. Between-subject grasp similarity was the similarity between the medoid grasp of each participant and the medoid grasp across all other participants.

## Normative model

The model takes as input 3D meshes of the stimuli and outputs a cost function describing the costs associated with every possible combination of finger and thumb position on the

accessible surface locations of our objects (i.e., those not in contact with the table plane). First, we define the center of each triangle in the mesh as a potential contact point. Then, given all possible combinations of thumb and index finger contact points $\overrightarrow{CP_T} = [x_T, y_T, z_T]$; $\overrightarrow{CP_I} = [x_I, y_I, z_I]$, the surface normal at both contact points $\overrightarrow{n_T} = [x_T^n, y_T^n, z_T^n]$; $\overrightarrow{n_I} = [x_I^n, y_I^n, z_I^n]$, and the CoM of the object $\overrightarrow{CoM} = [x_{CoM}, y_{CoM}, z_{CoM}]$, the five penalty functions we combined into a normative model of grasp selection were defined as follows:

**Force closure.**  For two-digit grasping, a grasp fulfills force closure when the grasp axis connecting thumb and index contact points lies within the friction cones resulting from the friction coefficient between object and digits [17]. A grasp that does not fulfill force closure will not be able to lift and freely manipulate the object, no matter the amount of force applied at the fingertips. A grasp perfectly fulfills force closure when the grasp axis is perfectly aligned with the vectors along which gripping forces are applied, which are the opposite of the contact-point surface normals. Therefore, we defined the force closure penalty function as the sum of the angular deviances (computed using the atan2 function) of the grasp axis from both force vectors $\overrightarrow{F_T} = -\overrightarrow{n_T}$; $\overrightarrow{F_I} = -\overrightarrow{n_I}$:

$$P_{FC}(\overrightarrow{CP_T}, \overrightarrow{CP_I}) = atan2(\|\overrightarrow{F_T} \times (\overrightarrow{CP_I} - \overrightarrow{CP_T})\|, \overrightarrow{F_T} \cdot (\overrightarrow{CP_I} - \overrightarrow{CP_T}))$$
$$+ atan2(\|\overrightarrow{F_I} \times (\overrightarrow{CP_T} - \overrightarrow{CP_I})\|, \overrightarrow{F_I} \cdot (\overrightarrow{CP_T} - \overrightarrow{CP_I}))$$

**Torque.**  If a force is applied at some position away from the CoM, the object will tend to rotate due to torque, given by the cross product of force vector and lever arm (the vector connecting CoM to the point of force application). Under the assumption that is possible to apply forces at the thumb and index contact points that counteract the force of gravity $\overrightarrow{F_g}$, we can compute the total torque of a grip as the sum of torques exerted by each contact point. Therefore, we defined the torque penalty function as the magnitude of the total torque exerted by a grip:

$$P_T(\overrightarrow{CP_T}, \overrightarrow{CP_I}) = \|(\overrightarrow{CoM} - \overrightarrow{CP_T}) \times \overrightarrow{-F_g} + (\overrightarrow{CoM} - \overrightarrow{CP_I}) \times \overrightarrow{-F_g}\|$$

**Natural grasp axis.**  Schot, Brenner, and Smeets [24] have carefully mapped out how human participants grasp spheres placed at different positions throughout the peripersonal space, and provide a regression model that determines the naturally preferred posture of the arm when grasping a sphere. We input the configuration of our current experimental setup into the regression model developed by these authors, and found the natural grasp axis for our participants to be $\overrightarrow{NGA} = [0.49\ 0.87\ 0]$. We therefore defined the natural grasp axis penalty function as the angular deviance from this established natural grasp axis:

$$P_{NGA}(\overrightarrow{CP_T}, \overrightarrow{CP_I}) = atan2(\|\overrightarrow{NGA} \times (\overrightarrow{CP_I} - \overrightarrow{CP_T})\|, \overrightarrow{NGA} \cdot (\overrightarrow{CP_I} - \overrightarrow{CP_T}))$$

**Optimal grasp aperture for precision grip.**  Cesari and Newell [26] have shown that, when free to employ any multi-digit grasp, human participants selected precision grip grasps only for cubes smaller than 2.5 cm in length. As cube size increases, humans progressively increase the number of digits employed in a grasp. Therefore, since our participants were instructed only to employ precision grip grasps, we defined the optimal grasp aperture penalty function as 0 for grasp sizes smaller than 2.5 cm, and as a linearly increasing penalty for grasp

sizes larger than 2.5 cm:

$$P_{OGA}(\overrightarrow{CP_T}, \overrightarrow{CP_I}) = \begin{cases} 0, & if \; \|\overrightarrow{CP_I} - \overrightarrow{CP_T}\| < 25mm \\ \|\overrightarrow{CP_I} - \overrightarrow{CP_T}\| - 25, & if \; \|\overrightarrow{CP_I} - \overrightarrow{CP_T}\| > 25 \; mm \end{cases}$$

In pilot work, we observed that a penalty map linearly increasing from 0 cm worked equally as well as one linearly increasing from 2.5 cm. In Experiment 3 we further observed that increasing this threshold up to 10 cm did not hinder model performance. However, constructing this penalty function with the 2.5 cm threshold motivated by previous literature will allow us, in future work, to construct penalty functions with multiple thresholds for multi-digit grasping, as those observed by Cesari and Newell [26].

**Object visibility.**   Under the assumption that humans are attempting to minimize the portion of the objects hidden from view by their hand, we defined the optimal visibility penalty function as the proportion of object still visible during each possible grasp. We first defined the line on the XZ plane that passes through the thumb and index finger contact points. We made the simplifying assumption that, given all possible surface points on the object $SP_{TOT}$, the surface points $SP_{OCC}(\overrightarrow{CP_T}, \overrightarrow{CP_I})$ that fall to the side of the line where the hand is located will be occluded. Therefore, the object visibility penalty function was defined as:

$$P_{OGA}\left(\overrightarrow{CP_T}, \overrightarrow{CP_I}\right) = \frac{Length(SP_{OCC}(\overrightarrow{CP_T}, \overrightarrow{CP_I}))}{Length(SP_{TOT})}$$

**Overall grasp penalty function.**   To obtain the overall grasp penalty function, each grasp penalty function was first normalized to the [0 1] range (i.e., across all possible grasps for each given object, independently of the other objects). Then, we took the sum of the individual penalty functions:

$$\begin{aligned} P_O(\overrightarrow{CP_T}, \overrightarrow{CP_I}) \\ = P_{FC}(\overrightarrow{CP_T}, \overrightarrow{CP_I}) + P_T(\overrightarrow{CP_T}, \overrightarrow{CP_I}) + P_{NGA}(\overrightarrow{CP_T}, \overrightarrow{CP_I}) + P_{OGA}(\overrightarrow{CP_T}, \overrightarrow{CP_I}) \\ + P_{RT}(\overrightarrow{CP_T}, \overrightarrow{CP_I}) \end{aligned}$$

For display purposes this final function was normalized to the [0 1] range. The minima of this overall grasp penalty function represent the set of grasps that best satisfy the largest number of constraints at the same time.

**Model fitting.**   In both Experiments 1 and 2, human participants executed repeated grasps to the same objects at each orientation. To fit the overall grasp penalty function to these human data, for each participant in each condition we first defined a human grasp penalty function $P_H(\overrightarrow{CP_T}, \overrightarrow{CP_I})$ in which all grasps selected by a participant onto an object were set to have 0 penalty, and all grasps that had not been selected were set to have a penalty of 1. Then, we fit the function:

$$P_{O,fit}(\overrightarrow{CP_T}, \overrightarrow{CP_I}) = \sqrt{\sum_i w_i * P_i(\overrightarrow{CP_T}, \overrightarrow{CP_I})^2}$$

to the human grasp penalty function. More specifically, we employed a nonlinear least-squares solver to search for the set of weights $w_i = [w_{FC}; w_T; w_{NGA}; w_{OGA}; w_{RT}]$ that minimized the

function:

$$F(\boldsymbol{w_i}) = \sqrt{\boldsymbol{R(\overrightarrow{CP_T}, \overrightarrow{CP_I})}} * \left[ \sqrt{\sum_i w_i * \boldsymbol{P_i(\overrightarrow{CP_T}, \overrightarrow{CP_I})}^2} - \boldsymbol{P_H(\overrightarrow{CP_T}, \overrightarrow{CP_I})} \right]$$

i.e. we searched for the set of weights for which $\boldsymbol{P_{O,fit}}$ best approximated the human grasp penalty function $\boldsymbol{P_H}$. The solver employed the trust-region-reflective algorithm; we set the lower and upper bounds of the weights to be 0 and 1, and 0.2 as the starting value for all weights. The number of non-selected grasps with $\boldsymbol{P_H(\overrightarrow{CP_T}, \overrightarrow{CP_I})} = 1$ vastly outnumbered the few selected grasps for which $\boldsymbol{P_H(\overrightarrow{CP_T}, \overrightarrow{CP_I})} = 0$. To avoid overfitting the model to the regions of the grasp space where $\boldsymbol{P_H(\overrightarrow{CP_T}, \overrightarrow{CP_I})} = 1$, we designed $\boldsymbol{R(\overrightarrow{CP_T}, \overrightarrow{CP_I})}$ as a regularization function which served to give equal importance to high and low penalty grasps in the human grasp penalty function. Thus, for grasps where $\boldsymbol{P_H(\overrightarrow{CP_T}, \overrightarrow{CP_I})} = 0$, $\boldsymbol{R(\overrightarrow{CP_T}, \overrightarrow{CP_I})}$ was equal to the number of times the participant had selected that specific grasp. For grasps where $\boldsymbol{P_H(\overrightarrow{CP_T}, \overrightarrow{CP_I})} = 1$ instead, $\boldsymbol{R\left(\overrightarrow{CP_T}, \overrightarrow{CP_I}\right)} = \frac{N_{G,selected}}{N_{G,non-selected}}$; where $N_{G,selected}$ was the total number of grasps performed by the participant onto the object, and $N_{G,non-selected}$ was the total number of non-selected grasps within the grasp manifold. This way for both selected and non-selected grasp regions, the sum of $\boldsymbol{R(\overrightarrow{CP_T}, \overrightarrow{CP_I})}$ was $N_{G,selected}$, and both regions of grasp space were accounted for equally during the fitting.

**Predicting grasps.** The minima of both the equally weighted (non-fitted) and the fitted overall grasp penalty functions represent the set of grasps predicted to be optimal under the weighted linear combination of the five penalty functions included in our normative model. To visualize these predicted optimal grasps, we sampled them from the minima of the penalty functions. First, we removed all grasps with penalty values greater than the lower 0.1th percentile. This percentile value was selected to approximately match the proportion of grasp space actually covered by human grasps. The remaining grasps were therefore all optimal or near-optimal. From this subset, we then randomly selected (with replacement) a number of grasps equal to the number of grasps executed by the human participants. The probability with which any one grasp was selected was set to be 1 minus the grasp penalty, thus grasps with zero penalty had the highest probability of being selected. These sampled grasps can then be projected back onto the objects for visualization purposes (Figs 12A and 13A), or they can be directly compared to human grasps using the grasp similarity metric described above (Figs 12B, 12C and 13C).

## Supporting information

**S1 Fig. Human and model grasping patterns for Experiments 1 and 2.** Grasping patterns from human participants (left), unfitted model (middle), and fitted model (right). (a) Grasping patterns on wooden objects from Experiment 1. (b) Grasping patterns on mixed material objects from Experiment 2.
(PDF)

**S2 Fig. Pattern of empirical results from Experiments 1 and 2 recreated from simulating grasps from the fitted model.** Panels are the same as in Figs 3, 4 and 5 of the main manuscript, except that the data are simulated from the model. The grasp trajectories in panel (4b) are from the human data, to highlight how the model correctly reproduces the biases in human grasping patterns. Panel 5b is omitted since the model cannot learn to refine CoM estimates.
(PDF)

**S3 Fig. Location of the center of mass for the stimuli employed in Experiments 1 and 2.** The center of mass of the light wooden objects from Experiment 1 is shown as a black dot. The centers of mass for the heavy alternate and bipartite wood/brass objects from Experiment 2 are shown as red dots and squares respectively.
(PDF)

## Acknowledgments

The authors thank Dr. Karl Gegenfurtner for insightful feedback, as well as Dr. Melvyn Goodale and Dr. Jody Culham for helpful discussion and encouragement.

## Author Contributions

**Conceptualization:** Lina K. Klein, Guido Maiello, Vivian C. Paulun, Roland W. Fleming.

**Data curation:** Lina K. Klein, Guido Maiello.

**Formal analysis:** Lina K. Klein, Guido Maiello.

**Funding acquisition:** Guido Maiello, Roland W. Fleming.

**Investigation:** Lina K. Klein, Guido Maiello, Vivian C. Paulun, Roland W. Fleming.

**Methodology:** Lina K. Klein, Guido Maiello, Vivian C. Paulun, Roland W. Fleming.

**Project administration:** Lina K. Klein, Guido Maiello.

**Resources:** Roland W. Fleming.

**Software:** Lina K. Klein, Guido Maiello, Vivian C. Paulun, Roland W. Fleming.

**Supervision:** Guido Maiello, Roland W. Fleming.

**Validation:** Lina K. Klein, Guido Maiello.

**Visualization:** Lina K. Klein, Guido Maiello.

**Writing – original draft:** Lina K. Klein, Guido Maiello, Roland W. Fleming.

**Writing – review & editing:** Lina K. Klein, Guido Maiello, Vivian C. Paulun, Roland W. Fleming.

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
