## [Decision Letter · Decision Letter 0]

15 Aug 2019

Dear Dr Maiello,

Thank you very much for submitting your manuscript 'Predicting precision grip grasp locations on three-dimensional objects' for review by PLOS Computational Biology. Your manuscript has been fully evaluated by the PLOS Computational Biology editorial team and in this case also by independent peer reviewers. The reviewers appreciated the attention to an important problem, but raised some substantial concerns about the manuscript as it currently stands. While your manuscript cannot be accepted in its present form, we are willing to consider a revised version in which the issues raised by the reviewers have been adequately addressed. We cannot, of course, promise publication at that time.

Sincerely,

Wolfgang Einhäuser

Deputy Editor

PLOS Computational Biology

[LINK]

Reviewer's Responses to Questions

**Comments to the Authors:**

Reviewer #1: The manuscript addresses an important topic and it provides an interesting development of the current views about how people select the contact points when they have to grasp objects. The authors acknowledge that this is an initial attempt at addressing this topic and it thus doesn't capture yet the full spectrum of possible grasping actions (e.g., multi-digit grasping, grasping complex shapes with curved surfaces, etc.). Nevertheless, it is a good contribution to this topic. I also had the chance to read the opinions of the previous reviewers who raised a number of important points which were successfully addressed by the authors.

Here below I'll list some elements that I think still need to be addressed.

line 254, Optimal grasp aperture: the authors say that 2.5 cm is the preferred distance between finger and thumb for precision grips citing Cesari and Newell, but in the present case the participants were explicitly instructed to use a precision grip, so I don't see why the 2.5 cm were chosen as an important value to define this penalty function. It would have made more sense to measure the maximum comfortable grasp aperture span of each participant and use those values to define the range of possible grasp apertures. Any grasp aperture between 0 and the maximum comfortable grasp aperture span should have a penalty function of 0. How robust is the computational model to different versions of the optimal grasp aperture penalty function such as the one I have mentioned above?

One aspect that is not mentioned in the manuscript is the fact that participants might have actually learned about the properties of the object during an experimental session. Did contact point selection change over the duration of an experimental session when participants had the chance to interact multiple times with the various objects and thus learn the characteristics of each object? For example, are the contact point locations more clustered if you compare the last trials with the first trials?

Line 215: The authors say that they do not suggest that the human brain explicitly evaluates grasp cost for all possible locations. This makes a lot of sense. However, another aspect that is also not very realistic is that humans actually have a VERIDICAL representation of the 3D object shape, its mass distribution, orientation and position that would be needed to even start any process similar to the one suggested by the authors. It would be important to mention also this aspect and it would be interesting to implement some perturbations to the inputs of the computation model and observe how robust the model is to them. For example, the 3D description of the shape could be slightly distorted, the mass distribution could not be perfectly accurate, the orientation and position slightly off: if the model is then still able to decently match human performance it would add value to the suggested model.

Figure S2: I would like the authors to speculate also about some surprisingly bad predictions of their model. For example, in Figure S2 (a), line 4, the Fitted Model predicts some grasps which are opposite to those in Human (thumb is predicted to land where humans placed the index, and the index is predicted to land where the humans placed the thumb). Or, in line 6, there is a cluster of index contact points on the lower right side that doesn't exist in human data and it seems to lack its corresponding thumb contact points (or are the predicted grasps here actually on the two sides of the corner?).

Typos:

line 215, typo: "are be optimal"

line 216, unclear: "It provides a means of based on a subset..."

line 454: though should be through

Figure S3, (a): "2 mode" should be "2 modes".

Reviewer #2: This manuscript describes sets of human-subjects experiments that investigate how people visually plan and grasp six differently shaped and oriented objects, using line of sight observation tracking to determine the positioning of the index finger and thumb. Based on this data, the results indicate high similarly between subjects, within an object. Based on the tracking data, a data-driven model involving several parameters is built to assess the relative importance of each (force closure, torque, axis, etc.) on grasp selection.

Overall, the experiments are well planned and conducted, and their resultant data are rigorously analyzed. The question of the visual planning of initial grasping location based on visual cues of an objects geometry, orientation, etc. is an interesting one, and builds upon prior work in this area. A model that seeks to unify the data from various studies could be a nice addition to the literature.

Major comments:

1. The experiments do leave one with a clear sense that particular grasp elements are more important than others. On the other hand, while the model seems to be the major contribution of this work, it is preliminary and not well enough articulated. First, the model is built up from observations with a set of objects and then used with those same set of objects, so one wonders how well is applies to a new set of objects or new case. Therefore, it seems like it well describes the observed phenomena from the human subjects experiments, but it's not clear how predictive it is of new situations/objects. This is problematic because most of the introduction and other sections denote it’s predictive ability. Second, even within what it does, I find its articulation (mostly done through Figure 4) hard to follow, and wonder if the authors could focus more on a single concrete example, of greater detail. One of the selling points of the MS is that there is a unified model that brings together findings from theirs and other studies, but if this is the central contribution, more time, space and figures should be spent on it, and demonstrating it. And I also wonder if the authors should generate new objects to determine how well it predicts grasp for those.

2. The current introduction is not well motivating the study. First, it's not precise in several regards surrounding "computation," as well as in whether grasp selection is initial selection or later selection and if feed-forward information can be employed because the object has now been touched several times, etc. Once you get to the section in the lines 50-65 range, you figure out where the MS is positioning itself, but before that point, I'm wondering where this manuscript is headed. The language and writing is quite loose. Second, all through reading that part, I'm asking myself these questions because of my familiarity with seminal work by Roland Johansson, and I'm wondering when I will hear about their work relative to it. The authors bring up Johansson, but not until the Discussion. After reading the whole MS, I do recognize what they're doing is different. However, because that Johansson work is so prominent, the introduction must address it.

3. Several key terms are either unclearly defined or misused. The authors use the term "computational model" in many places, and many readers (including me) would might say this is not a computational model. It is essentially a data-driven, regression-like model, that is a not a physics based, but is built up from observations of people interacting with objects. They also use the term "predict" but they do not per se predict how well the model can be applied in using new objects, as previously noted. It works well with objects it was built around, and perhaps it could do well for new objects, but that's not shown. Later in the manuscript, the term "normative" model is used. The authors need to fix the "computational" and "predict" terminology to be accurate with what was done. Another term used in the abstract and various other places regards material properties, but the objects used in these studies rigid bodies with flat, smooth surfaces, so the material properties are not varied. It could be however, that the wood and brass do present different coefficients of friction, which might be problematic. Finally, it would be good if the authors were more clear that they are looking at the "initial grasp position" rather than the "readjusted during the grasp position". And their force closure is more simplistic than the grip and lift forces that Johansson describes.

4. Like Fig. 4, but more to a point of formatting, I think Fig. 3 (described below) should be re-formatted or broken up into distinct figures. At present, there are too many sub-figures, with too small of text, of unclear relationship to one another.

Detailed comments:

A. Abstract, Last sentence: "Together, the findings provide a unified account of how we successfully grasp objects with different 3D shapes and material properties." - The authors mention material properties. The other factors mentioned are shape, weight, orientation, and mass distribution but these are not material properties, per se as one would view from a mechanical engineering perspective. Material properties are elasticity, and/or it's time dependent nature, hysteresis, and surface texture, etc. This should be clarified.

B. Intro, about lines 36-48. Moreover, Westling and Johansson, 1984, "Roles of glabrous skin receptors and sensorimotor memory in automatic control of precision grip when lifting rougher or more slippery objects" have done extensive work on grasp, and looked at the ratio of lift to grip forces in picking up various objects and how people use safety factors to ensure the object does not slip on them, they also considered the use of feed-forward visual estimation that drives selection of the appropriate grasp, and in their experiments varied the surface contact interactions by using various materials, from sand paper to slippery velvet, etc. This paper and other papers of theirs have been cited thousands of times, and are of mainstream thought in the field of haptics. The MS does cite 3 Johansson papers but only in the discussion section.

C. Intro, Line 22, "computational complexity" seems to refer to how humans perform the task, or is it how AI performs the task? It should be clarified as to if the authors are implying that humans are using computational processes to make decisions about grasp and lift tasks, or if they are referring to sensors and actuators on robots and the associated computational and/or analytical algorithms used to inform grip adjustment. This terminology is also used in Figure 1's caption and elsewhere.

D. Intro, lines 49-50, "we sought to unify these varied and fragmented findings into a single computational framework." This is not really a hypothesis, per se. What comes after this mentions constructing a dataset and teaching apart the five factors. And then onto the factors varied per experiment, but this writing is fairly loose and not tied tightly enough to a precise goal and to the prior literature. After reading this section, I'm struggling to determine the direction of this work, especially without any tie to the seminal Johansson references.

E. Intro, lines around 45, In this paragraph, sometimes the language is from the object's perspective (e.g., mass, mass distrib, shape) and others from the person's grasp (orientation, size). Make it consistent for the case being described.

F. Intro, lines around 45, In this paragraph, since CoM may be hard to interpret visually for some strange geometrics, are we just restricting the grasp to the initial grasp? Or readjusted in mid-lift? Also, the second time the participants have seen it, they would have some feed-forward information. Would they have adjusted their grasp the second time they saw a configuration? Was this last point analyzed?

G. It is very difficult in figure 3 to make sense of how all of these figures tie together. Some have aggregate data for all participants, some appear to be for one or another factor or case, etc. What is the story to be told by these? Figure 3 should be reformatted or broken into more than one figure. Figure, 3b - what are the axis units and labels? Figure 3f - How does this figure tie to the others? And is this just an example figure? How is the 26 mm noted to be gleaned from Figure 3f? The figure is quite small and it is unclear if the axes represent quantities to scale.

H. Intro, line 65, a model is mentioned. At this point, the MS needs to be more specific on what are the input-output relationships and what type of model (regression, free-body, dynamics, solid mechanics, etc.). I think it would also be good to be clear that w.r.t. force closure and torque that these are acquired by free-body physics, and involve no direct measurements of force related quantities which might come from sensors on the device itself. I’m not sure how you can get actual torque if you do not have normal force and are not tracking the movement of the block pattern.

J. Results, line 156-7. "...our findings suggest humans care little about torque when grasping lightweight objects." This statement should be quantified, if left in results, or else moved to methods if meant to be brought up in a higher-level, abstract way. What torque range are we referring? What range of weights of objects are we referring?

K. Results, in several points between lines 160-195. It seems like the participants are estimating CoM by their own volition. They were not taught how to do this, but perhaps they make guesses in their head as to the configurations of the wood and brass blocks, and resulting CoM? Did you consider painting these so that participants would not know the composition of block patterns that they're pickup up? Perhaps they participants when doing this task see the metals, think they might be heavier regardless of configuration, and therefore they choose a safer position, rather than faster/easier in the all wood case.

L. Results, lines 241-265. This section discusses the metrics and penalty function calculations. Instead of talking about these in a general way, why not tie these directly to Figure 4. The rest of the paragraphs before and after do just that. In this way, the model description could be strengthened.

M. Lines 392-3, "... can be applied to any set of novel objects ..." Yet this was not done. If the model is to be predictive, as described at various points in the MS, it needs to do just this.

N. Lines 423-439, Section "Neuroscience of Grasping" should be moved to the discussion or eliminated. This does not belong in the results section as there are no results here, nor in methods.

O. Methods, line 443. Can you be clear that there was no overlap between the subjects who were involved in Exps. 1 & 2?

P. Methods, Apparatus, line 460. The Optotrak 3020 was used. This system (as I find in the on-line manuals) indicates a 0.1 mm 3d accuracy and 0.01 mm resolution. This seems quite good and suitable for your purposes. But it reads "... up to ..." Did you test what indeed the system could achieve? Is your separation between different measurements above these levels? It should at least be specified what are the device specs.

Q. Methods, lines 522. About 14% of trials were removed due to some form of error. Of those, more than 90% were due to tracking problems. Could it be that these are valid grasp strategies and common ones at that, but that the line of sight of the camera could not capture the necessary angles?

Reviewer #3: This is an elegant and new account of how humans choose grasping locations on 3D objects based on a simple normative theory. The theory is implemented by a computational model that combines five cost functions in order to locate on wooden block objects ideal precision grip locations. The model predicts quite well, even without free parameters, where humans place their index finger and thumb to firmly pick up these unfamiliar 3D objects.

I believe that the findings are new and interesting and that the manuscript may deserve to be published. I have, however, concerns that need to be addressed before the manuscript can be accepted for publication.

First, I wander how generalizable the model is beyond predicting grasp locations on “cube objects”. Given the quite obvious assumptions of force closure and natural axis, the four objects tested do not offer a large number of alternative locations for stable grasps. Also, the constraint of limiting the grasps to 2.5cm, although based on previous studies, seems almost as it had been chosen after the empirical data collection, since subjects clearly preferred smaller grips. My point being, it looks like a quick inspection of the data about grip location selection presented in the supplementary material already shows evidence of what constraints humans use to select their grip, which makes the development of a computational model perhaps redundant.

Second, it seems to me that in Experiment 2 subjects pick-up the objects by basically ignoring the wooden parts, given how much heavier the brass parts are in comparison. Not surprisingly, the model does the same since the CoM is almost entirely determined by the brass parts. This is especially evident in the “near” and “far” conditions.

Third, the model is not based on an analysis of the 3D shape per se, since the input is a series of discrete grasping locations, which the model analyzes in a pair-wise fashion. If I understand the model correctly then I imagine there are many scrambled versions of the objects that would elicit the same predictions.

I summary, I think the model does a good job in predicting the empirical data. However, given the small set of objects tested and the reduced set of possible grasping locations that these objects offer, I am not sure what the model adds to what is already evident from a careful inspection of the raw data.

**Have all data underlying the figures and results presented in the manuscript been provided?**

Reviewer #1: No: The authors wrote that data and analysis scripts will be provided upon publication.

Reviewer #2: None

Reviewer #3: Yes

PLOS authors have the option to publish the peer review history of their article (what does this mean?). If published, this will include your full peer review and any attached files.

Reviewer #1: No

Reviewer #2: No

Reviewer #3: No

---

## [Decision Letter · Decision Letter 1]

21 May 2020

Dear Ms. Klein,

Thank you very much for submitting your manuscript "Predicting precision grip grasp locations on three-dimensional objects" for consideration at PLOS Computational Biology. As with all papers reviewed by the journal, your manuscript was reviewed by members of the editorial board and by several independent reviewers. The reviewers appreciated the attention to an important topic. Based on the reviews, we are likely to accept this manuscript for publication, providing that you modify the manuscript according to the review recommendations.

As you will see in their comments below, the reviewers are now satisfied with your manuscript. Reviewer 1 has one further suggestion, which I would like to ask you to consider - so I set the decision to minor revision to give you the opportunity to include this. Once this is done, the manuscript will be ready for acceptance without further rounds of review. Please also include the link to your data (the doi provided by the archive) in the final manuscript.

Sincerely,

Wolfgang Einhäuser

Deputy Editor

PLOS Computational Biology

[LINK]

As you will see in their comments below, the reviewers are now satisfied with your manuscript. Reviewer 1 has one further suggestion, which I would like to ask you to consider - so I set the decision to minor revision to give you the opportunity to include this. Once this is done, the manuscript will be ready for acceptance without further rounds of review.

Reviewer's Responses to Questions

**Comments to the Authors:**

Reviewer #1: The authors have successfully addressed all the concerns I have raised in my previous review. I only have one additional minor comment:

The current implementation of the model is not considering that some parts of the object could act as obstacles. This would explain some of the patterns that can be observed in Figure S1(a). For example, when the S and V objects are in the S1, S2 and V1 orientations, there is one part of the object that is sticking up and thus obstructs possible grasps of the parts hidden behind it. Humans clearly take this aspect into account (they rarely moved the hand over the obstacle to grasp a part behind it). On the other hand, both the Unfitted and the Fitted models disregard the sticking part and choose grasps that differ from those made by humans. Interestingly, in S1(b) humans are now actually grasping over the obstacle because the cost of doing it is probably less than the cost of grasping the object to far away from the COM. The model could thus incorporate also a component that takes into account how long (or curved) the movements are. If a part of the object obstructs a direct reaching movement, the movement should be necessarily longer (more curved) and thus have a higher cost. I would say that at this stage is up to the authors to include this component into the model or just mention it as a possible limitation of the model.

Reviewer #3: I am pleased with how the authors addressed my previous concerns. In particular, I am glad the model is able to generalize the predictions to novel objects that are substantially different from polycube objects made of brass and wood.

**Have all data underlying the figures and results presented in the manuscript been provided?**

Reviewer #1: No: The authors state that the data and analysis scripts will be available in a Zenodo database upon publication, so they are currently not provided.

Reviewer #3: Yes

PLOS authors have the option to publish the peer review history of their article (what does this mean?). If published, this will include your full peer review and any attached files.

Reviewer #1: No

Reviewer #3: No
---

## [Editor Report · Decision Letter 2]

22 Jun 2020

Dear Dr. Maiello,

We are pleased to inform you that your manuscript 'Predicting precision grip grasp locations on three-dimensional objects' has been provisionally accepted for publication in PLOS Computational Biology.

Best regards,

Wolfgang Einhäuser

Deputy Editor

PLOS Computational Biology

---

## [Editor Report · Acceptance letter]

27 Jul 2020

PCOMPBIOL-D-19-00826R2 

Predicting precision grip grasp locations on three-dimensional objects

Dear Dr Maiello,

I am pleased to inform you that your manuscript has been formally accepted for publication in PLOS Computational Biology. Your manuscript is now with our production department and you will be notified of the publication date in due course.

With kind regards,

Matt Lyles
